# Increased carbon capture by a silicate-treated forested watershed affected by acid deposition

Lyla L. Taylor[1*], Charles T. Driscoll[2], Peter M. Groffman[3], Greg H. Rau[4], Joel D. Blum[5] and David J. Beerling[1]

[1]Leverhulme Centre for Climate Change Mitigation, Department of Animal and Plant Sciences, University of Sheffield, Sheffield S10 2TN, UK

[2]Department of Civil and Environmental Engineering, 151 Link Hall, Syracuse University, Syracuse, NY 13244, USA

[3]City University of New York, Advanced Science Research Center at the Graduate Center, New York, NY 10031 and Cary Institute of Ecosystem Studies, Millbook, NY 12545 USA

[4]Institute of Marine Sciences, University of California, Santa Cruz, CA 95064 USA

[5]Department of Earth and Environmental Sciences, University of Michigan, Ann Arbor, MI 48109, USA

*Correspondence to*: Lyla L. Taylor (L.L.Taylor@sheffield.ac.uk)

**Abstract.** Meeting internationally agreed-upon climate targets requires Carbon Dioxide Removal (CDR) strategies coupled with an urgent phase-down of fossil fuel emissions. However, the efficacy and wider impacts of CDR are poorly understood. Enhanced rock weathering (ERW) is a land-based CDR strategy requiring large-scale field trials. Here we show that a low 3.44 t ha$^{-1}$ wollastonite treatment in an 11.8-ha acid-rain-impacted forested watershed in New Hampshire, USA led to cumulative carbon capture by carbonic acid weathering of 0.025–0.13 t $CO_2$ ha$^{-1}$ over 15 years. Despite a 0.8–2.4 t $CO_2$ ha$^{-1}$ logistical carbon penalty from mining, grinding, transportation and spreading, by 2015 weathering together with increased forest productivity led to net CDR of 8.5–11.5 t $CO_2$ ha$^{-1}$. Our results demonstrate that ERW may be an effective, scalable CDR strategy for acid-impacted forests but at large-scale requires sustainable sources of silicate rock dust.

## 1 Introduction

The Intergovernmental Panel on Climate Change (IPCC)(Rogelj et al., 2018) Special Report on global warming indicates large-scale deployment of Carbon Dioxide Removal (CDR) technologies will be required to avoid warming in excess of 1.5 °C by the end of this century. Land-based CDR strategies include enhanced rock weathering (ERW), which aims to accelerate the natural geological process of carbon sequestration by amending soils with crushed reactive calcium (Ca) and magnesium (Mg)-bearing rocks such as basalt (The Royal Society and The Royal Academy of Engineering, 2018;Hartmann et al., 2013). Forests represent potential large-scale deployment opportunities where rock amendments may provide a range of benefits, including amelioration of soil acidification and provisioning of inorganic plant-nutrients to cation-depleted soils (Hartmann et al., 2013;Beerling et al., 2018). Although ERW has not yet been demonstrated as a CDR technique at the catchment scale, a forested watershed experiment at the Hubbard Brook Experimental Forest (HBEF, 43° 56'N, 71° 45'W) in the White

Mountains of New Hampshire, USA provides an unusual opportunity for assessing proof-of-concept in this priority research area.

The HBEF watershed experiment, designed to restore soil calcium following decades of leaching by acid rain, involved application of a finely ground rapidly-weathered calcium silicate mineral wollastonite ($CaSiO_3$; 3.44 t ha$^{-1}$) on 19 October 1999 to an 11.8-ha forested watershed (SI Appendix) (Likens et al., 2004;Peters et al., 2004;Shao et al., 2016).  Unlike the carbonate minerals (e.g., $CaCO_3$) commonly applied to acidified soils (Lundström et al., 2003), wollastonite does not release $CO_2$ when weathered (Supplementary Information) so is much better suited for CDR (Hartmann et al., 2013).  It also has dissolution kinetics comparable to or faster than other calcium-rich silicate minerals such as labradorite found in basalt (Brantley et al., 2008).   Thus, the HBEF experiment provides a timely and unparalleled opportunity for investigating the long-term (15 years) effects of ERW on CDR potential via forest and stream water chemistry responses.

In the case of ERW with wollastonite, CDR follows as Ca cations ($Ca^{2+}$) liberated by weathering consume atmospheric $CO_2$ through the formation of bicarbonate ($HCO_3^-$) by charge balance, as described by the following reaction:

$$CaSiO_3 + 3H_2O + 2CO_2 \rightarrow Ca^{2+} + H_4SiO_4 + 2HCO_3^-, \hspace{4cm} (R1)$$

However, forests in the northeastern USA have experienced acid deposition (Likens and Bailey, 2014), changes in nitrogen cycling (Goodale and Aber, 2001;McLauchlan et al., 2007) and increases in dissolved organic carbon (DOC) fluxes (Cawley et al., 2014) that may affect $CO_2$ removal efficiency by ERW processes.  In particular, $CO_2$ consumption as measured by bicarbonate production may be diminished if sulphate ($SO_4^{2-}$), nitrate ($NO_3^-$), or naturally-occurring organic acid anions (Fakhraei and Driscoll, 2015) ($H_2A^-$) in DOC intervene to inhibit the following mineral weathering reactions. For example:

$$CaSiO_3 + H_2O + H_2SO_4 \rightarrow Ca^{2+} + H_4SiO_4 + SO_4^{2-}, \hspace{4cm} (R2)$$

$$CaSiO_3 + H_2O + 2HNO_3 \rightarrow Ca^{2+} + H_4SiO_4 + 2NO_3^-, \hspace{4cm} (R3)$$

$$CaSiO_3 + H_2O + 2H_3A \rightarrow Ca^{2+} + H_4SiO_4 + 2H_2A^-, \hspace{4cm} (R4)$$

These environmental effects on stream-water chemistry are well documented at the HBEF (Cawley et al., 2014;Likens and Bailey, 2014;Rosi-Marshall et al., 2016;McLauchlan et al., 2007), and may be exacerbated under future climate change (Sebestyen et al., 2009;Campbell et al., 2009).

Here we exploit the experimental design and long-term monitoring of streamwater chemistry, trees, and soils, for two small forested HBEF watersheds to evaluate the effects of the wollastonite treatment in 1999 on catchment $CO_2$ consumption via inorganic and organic pathways.  Further, we examine how biogeochemical perturbations in S, N, and organic carbon cycling affect catchment inorganic $CO_2$ consumption.  We consider the forest response, the carbon cost for ERW deployment (mining, grinding, transportation and application), and the net greenhouse gas balance for the treatment.  Finally, we provide an initial assessment of the net CDR potential of silicate treatments deployed over larger areas of acidified forest in the northeastern United States.

## 2 Methods

This section describes the site and wollastonite treatment (Section 2.1) and our approaches for modelling the inorganic carbon fluxes in streamwater (Section 2.2) and other greenhouse gas fluxes associated with the treatment (Section 2.3). The variables from each of the seven equations in Methods are tabulated in Table 1 along with the section, equation, figure and table numbers where they appear.

**Table 1. Summary of variables presented in Methods (Section 2, Eqs 1 through 7)**

| Variable | Units | Sections | Equations | Figures | Tables | Description |
|---|---|---|---|---|---|---|
| $[HCO_3^-]$ | mol kgw$^{-1}$ | 2.2.3, 2.2.4 | 1,3 | 2a, S5a,d,g S6a | | Concentrations of solutes in water, in this case $HCO_3^-$, are denoted by square brackets. |
| $t$ | time | 2.2.3–5 | 1–5 | | | Denotes individual samples in the time series |
| $\alpha_{rain,HCO3}$ | fraction | 2.2.3, 2.2.4, Appendix A | 1,2,A2 | S3 | | Fraction of an ion, in this case $HCO_3^-$, originating from precipitation. |
| $flow$ | mm year$^{-1}$ | 2.2.3–5 | 1–5 | S2d | | Streamwater flow |
| $CO_{2,HCO3}$ | mol C year$^{-1}$ | 2.2.3, 2.2.4 | 1,3 | 2b, S5b,e,h | 3,S1 | Total watershed $CO_2$ consumption as calculated from bicarbonate |
| $CO_{2,ions}$ | mol C year$^{-1}$ | 2.2.3 | 2 | | 3 | Total watershed $CO_2$ consumption as calculated from major ions |
| $Wo\text{-}CO_{2,HCO3}$ | mol C year$^{-1}$ | 2.2.4 | 3 | 2c,5, S5c,f,i S6b,c | 3,4,S1 | Watershed $CO_2$ consumption as calculated from bicarbonate resulting from wollastonite weathering. Our conservative/pessimistic $\Delta CONS$ estimate in our GHG balance is the 15-year sum. |
| $X_{Ca}$ | fraction | 2.2.3, Appendix B | 3,4,B1 | S1a | | Fraction of total calcium originating from wollastonite |
| $Wo\text{-}CO_{2,Ca}$ | mol C year$^{-1}$ | 2.2.3 | 4 | 2f, S6b,c | 3, S1 | Watershed $CO_2$ consumption as calculated from calcium due to wollastonite weathering. Our optimistic estimate for $\Delta CONS$ in our GHG balance is the 15-year sum. |
| $C_i$ | mol kgw$^{-1}$ | 2.2.5 | 5 | | | Concentration of solute for sample $i$ (collected ~monthly for chemical analysis) |
| $Q_i$ | mm time$^{-1}$ | 2.2.5 | 5 | | | Streamflow for sample $i$ |
| $Q_k$ | mm day$^{-1}$ | 2.2.5 | 5 | S2b | | Streamflow for day $k$ |
| $N$ | number | 2.2.5 | 5 | | | Number of daily flow measurements |
| $\Delta GHG$ | t $CO_2$ ha$^{-1}$ | 2.3.1 | 6 | 5 | 4 | Net treatment effect on watershed greenhouse gas balance |
| $\Delta wood$ | t $CO_2$ ha$^{-1}$ | 2.3.1 | 6 | 5 | 4 | Treatment effect on woody biomass over ten years, positive if wood production increases relative to reference watershed. |
| $\Delta CH4$ | t $CO_2$ ha$^{-1}$ | 2.3.1 | 6 | 5 | 4 | Treatment effect on soil $CH_4$ sink since 2002, positive if the soil $CH_4$ sink increases relative to reference watershed. |

| | | | | | | |
|---|---|---|---|---|---|---|
| *ΔSRESP* | t $CO_2$ $ha^{-1}$ | 2.3.1 | 6 | 5 | 4 | Treatment effect on soil $CO_2$ emissions since 2002, positive if emissions decrease relative to reference watershed. |
| *ΔCONS* | t $CO_2$ $ha^{-1}$ | 2.3.1 | 6 | 5 | 4 | Treatment effect on $CO_2$ consumption over 15 years, range from **Wo-$CO_{2,HCO3}$** and **Wo-$CO_{2,Ca}$** |
| *ΔN2O* | t $CO_2$ $ha^{-1}$ | 2.3.1 | 6 | 5 | 4 | Treatment effect on soil $N_2O$ emissions since 2002, positive if emissions decrease relative to reference watershed. |
| *ΔNO3N2O* | t $CO_2$ $ha^{-1}$ | 2.3.1 | 6 | 5 | 4 | Treatment effect on downstream $N_2O$ emissions (due to nitrate export) over 15 years, positive if emissions decrease relative to reference watershed. |
| *ΔDOC* | t $CO_2$ $ha^{-1}$ | 2.3.1 | 6 | 5 | 4 | Treatment effect on dissolved organic carbon export over 15 years, positive if export decreases relative to reference watershed. This represents carbon loss from the watershed and likely $CO_2$ emissions downstream. |
| *LOGPEN* | t $CO_2$ $ha^{-1}$ | 2.3.1 | 6 | 5 | 4 | Logistical emissions penalty associated with mining, milling, pelletization, transport and application of the wollastonite treatment, expected to be negative. |
| *s* | $m^2$ $kg^{-1}$ | 2.3.3 | 7 | | | Specific surface area of material being milled |
| $e_p$ | kJ $kg^{-1}$ | 2.3.3 | 7 | | | Specific potential energy of material being milled |


 **2.1 Site and treatment**

 **2.1.1 Site description**

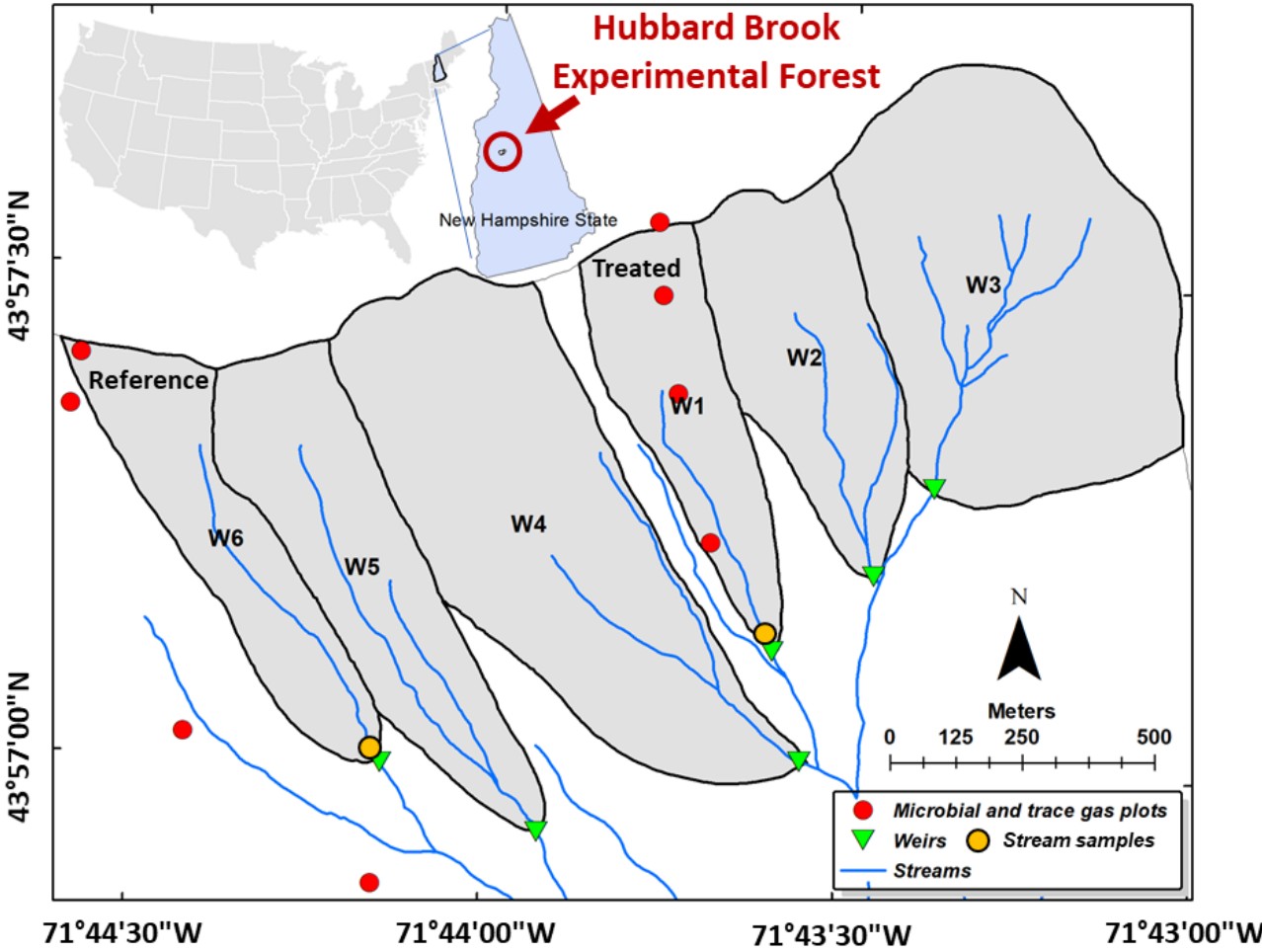

**Figure 1: Location of the sampling sites and experimental watersheds.** Our streamwater samples were collected just
upstream of the weirs in the treated and reference watersheds (gold disks) and our trace gas samples were collected at different
elevations in treated and untreated forests (red disks).
The HBEF has a temperate climate with ~1400 mm mean annual precipitation of which up to one third falls as snow (Campbell
et al., 2007). The mean temperatures in January and July are  –9 °C and 18 °C respectively, and the period from mid-May to
mid-September comprises the growing season (Campbell et al., 2007).  There are six small southeast-facing watersheds in the
HBEF (**Fig. 1**) with 20%–30% slopes (Groffman et al., 2006), including one which received the silicate treatment (watershed
W1, 11.8 ha, 488–747m asl) and a biogeochemical reference (watershed W6, 13.2 ha, 545–791m asl). Carbonate and evaporite
minerals are in very low abundance (<1% calcite in the crystalline rocks and glacial deposits) in these silicate-mineral
dominated watersheds (Johnson et al., 1981). Well-drained Typic Haplorthod soils with pH<4.5 and mean depth 0.6m formed
from relatively impermeable glacial till, which restricts water flow and protects the underlying schist bedrock from weathering.
Overland runoff and flow through bedrock are both thought to be negligible (Likens, 2013). Hydrologically, the HBEF
watersheds are typical of small catchments in northern New England (Sopper and Lull, 1965). Flow rates for W1 and W6
along with streamwater pH are shown in Fig. S1. Prior to treatment, streamwater calcium concentrations were under 30 μmol
$L^{-1}$ while bicarbonate concentrations were under 5 μmol $L^{-1}$, below the ranges for typical world rivers (Moon et al., 2014) (60–
2293 μmol $Ca^{2+}$ $L^{-1}$, 179–4926 μmol $HCO_3^-$ $L^{-1}$).
*Fagus grandifolia, Betula alleghentiensis* and *Acer saccharum* are the dominant trees in this Northern Hardwood forest,
while *Betula papyrifera, Abies balsamea* and *Picea rubens* are common at the highest elevations where soils tend to be shallow
and wetter (Cho et al., 2012). *A. saccharum* and *P. rubens* are both calcium-sensitive, but soil calcium-bearing minerals are
less available to *A. saccharum* (Blum et al., 2002) and total bioavailable calcium content decreases with elevation (Cho et al.,
2012). This silicate-addition experiment was designed to replace bioavailable calcium which had been stripped from the soils
by decades of acid deposition.
**2.1.2 Treatment description**
On 19 and 21 October 1999, W1 was treated with 344 $g/m^2$ of pelletized wollastonite ($CaSiO_3$) by a GPS-equipped helicopter
with a motorized spreader to ensure even deployment across the catchment, including the 1804 $m^2$ streambed (Peters et al.,
2004). Following treatment, the lignin-sulfonate binder forming the pellets dissolved within several days (Peters et al., 2004),
and the ground wollastonite itself dissolved rapidly in the upper Oie soil horizon, increasing Oie base saturation from 40% to
78% and raising soil pH from 3.88 to 4.39 within one year (Johnson et al., 2014). Although the budget of wollastonite-derived
calcium (Wo-Ca) has never been closed due to lack of data from vegetation and from deeper soil layers (Shao et al., 2016), it
is thought that uptake by vegetation and retention by soil exchange sites delayed transport of Wo-Ca to lower soil horizons
and streamwater for three years (Johnson et al., 2014).
**2.2 Geochemical modelling and $CO_2$ consumption fluxes**
$CO_2$ consumption, the CDR pathway most closely associated with ERW, can be calculated from concentrations of either
bicarbonate or the base cations released during weathering (Eq. R1). These two approaches may provide different answers if
bicarbonate is reduced in the presence of other acids (Eqs. R2–R4). To calculate bicarbonate-derived $CO_2$ consumption, we
must model the speciation of dissolved inorganic carbon (DIC). This depends on two variables which must also be modelled
because we do not have a time series: streamwater $pCO_2$ and streamwater temperature. We then calculate total catchment $CO_2$
consumption fluxes and treatment effects, taking care to account for differences in sampling frequency between chemistry
samples and water flow measurements.

## 2.2.1 Forward modelling of streamwater chemistry including dissolved inorganic carbon

We used a forward modelling approach to calculate dissolved streamwater bicarbonate concentrations ($[HCO_3^-]_{stream}$) in the treated and reference watersheds (**Fig. 1**) over ~25 years, including 15 years post-treatment, with the United States Geological Survey (USGS) aqueous geochemistry software PHREEQC version 3.3.12-12704 (Parkhurst and Appelo, 1999) and monthly long-term (1992–2014) streamwater (Driscoll, 2016b, a) and rain/snow precipitation (Likens, 2016b, a) chemistry measurements.

Using MATLAB (version R2016a) scripts, we wrote PHREEQC input files and determined the inorganic carbon species for each streamwater sample with PHREEQC. Along with a standard database which decouples ammonium and nitrate (Amm.dat, provided with the PHREEQC software), we included the ionization constants for the organic acid triprotic analogue and the constants for Al complexation described for Hubbard Brook streams (Fakhraei and Driscoll, 2015) in our PHREEQC simulations. These are: $pK_{a1}=2.02$, $pK_{a2}=6.63$, $pK_{a3}=7.30$, $pK_{Al1}=4.07$, $pK_{Al2}=7.37$, $pK_{Al3}=6.65$, and site density $m=0.064$ mol sites mol C$^{-1}$. Our organic acid concentrations are the product of the corresponding site density of reactions and the measured dissolved organic carbon concentration (Fakhraei and Driscoll, 2015); these were PHREEQC inputs along with total monomeric Al and major ion concentrations from the longitudinal datasets.

Spectator ions ($Cl^-$ and $NH_4^+$) were adjusted to achieve charge balance given the measured pH for the treated and reference watersheds. $Cl^-$ was only adjusted when charge balance was not achieved using $NH_4^+$ alone. This was deemed to be the case when PHREEQC failed to converge or when the percent error exceeded 5%. We used original rather than adjusted rainwater Cl to calculate the contribution of rainwater to streamwater chemistry (described below). These adjusted ions were then held constant for our modelled scenarios, while pH was allowed to vary.

Exploratory PHREEQC tests (charge-balancing on DIC) either with or without organic acids suggest that the acids depress total DIC, $HCO_3^-$ and also the saturation state of gaseous $CO_2$. Similar variability in the saturation is also observed when DIC values from partially degassed samples from the streams are used as input. We chose minimum and maximum values of 1100 and 1700 ppm, or ~3 and $4.6 \times 368$, the mean value of Mauna Loa $pCO_2$ (Tans and Keeling, 2017) for 1985–2012. These values correspond to $\log_{10}(pCO_2(g)) = -2.87\pm0.09$ SD derived from a prior analysis of this variability for the same time range (Fakhraei and Driscoll, 2015).

## 2.2.2 Streamwater temperature

Air temperatures for the Hubbard Brook watersheds (Campbell, 2016) were converted to streamwater temperatures (Mohseni and Stefan, 1999). Rainwater temperatures were set equal to streamwater temperatures. These temperatures were used in our PHREEQC modelling, with equilibrium constants for the DIC species as functions of temperature. Only samples measured closest to the weirs and with a valid pH were processed with PHREEQC.

**2.2.3 Total catchment $CO_2$ consumption**

We calculate total annual watershed $CO_2$ consumption (Eq. R1) as the product of streamwater flow and streamwater bicarbonate concentration $[HCO_3^-]$ at time $t$ corrected for the $HCO_3^-$ contribution from rainwater ($\alpha_{rain,HCO3}(t)$, Appendix A):

$$CO_{2,HCO3}(t) = (1-\alpha_{rain,HCO3}(t))\ [HCO_3^-](t) \times flow(t), \tag{1}$$

where $[HCO_3^-](t)$ is given in mol kgw$^{-1}$ and $flow(t)$ is the "runoff" in mm year$^{-1}$. Calculated $[HCO_3^-]$ and annual $CO_2$ consumption for the treated and reference watersheds (Eq.1) comprise our baseline simulations and represent a primary test of hypothesized increased carbon capture resulting from weathering of the applied silicate.

Bicarbonate-derived $CO_2$ consumption (Eq. 1) is the most conservative approach to estimating net carbon fluxes related to ERW. For natural freshwaters in equilibrium with the atmosphere, this entails a titration for total alkalinity with a possible correction for the concentration of organic acid anions (Köhler et al., 2000). However, another widely used (Jacobson and Blum, 2003) measure of $CO_2$ consumption is derived by assuming that any base cations ($Ca^{2+}$, $Mg^{2+}$, $K^+$ and $Na^+$) released from minerals will be charge-balanced by bicarbonate formation in the oceans

$$CO_{2,ions}(t) = (2[Ca^{2+}](1-\alpha_{rain,Ca}(t))+ 2[Mg^{2+}](1-\alpha_{rain,Mg}(t))+[K^+](1-\alpha_{rain,K}(t))+$$
$$[Na^+](1-\alpha_{rain,Na}(t)) -2[SO_4^{2-}](t)) \times flow(t), \tag{2}$$

where the term $-2[SO_4^{2-}](t)$ represents a commonly-applied correction for sulphuric acid weathering (Chetelat et al., 2008) (Eq. R2). Contributions from precipitation such as $\alpha_{rain,Ca}(t)$ are calculated by replacing bicarbonate with the individual base cation in Eq. (A2). We tabulate $CO_{2,ions}(t)$ results for comparison with $CO_{2,HCO3}(t)$ below.

**2.2.4 Response of $CO_2$ consumption to treatment**

To isolate a treatment effect for bicarbonate, we used strontium isotopes as a tracer of wollastonite (Wo) weathering within a previously-published mixing function (Nezat et al., 2010;Peters et al., 2004) (Appendix B, **Fig. S3**). This mixing function provides the fraction $X_{Ca}$ of calcium originating from wollastonite (Eq. B1). We remove the contribution of all mineral sources other than wollastonite to $CO_2$ consumption (Eq. 1), which is simulated with $Ca^{2+}$ concentrations reduced by $(1-X_{Ca})$:

$$Wo\text{-}CO_{2,HCO3}(t) = CO_{2,HCO3}(t) - \{[HCO_3^-](t,(1-X_{Ca})\,[Ca^{2+}]) \times (1-\alpha_{rain,HCO3}(t)) \times flow(t)\} \tag{3}$$

which effectively provides a lower limit on the treatment effect.

We can also derive an upper limit for the treatment effect from Eq. (R1). For an ERW treatment, transient changes in the export of ions not derived from the applied minerals may occur, but we consider that the cations released from the applied minerals comprise the most unambiguous treatment effect in our study. The charge associated with wollastonite-derived $Ca^{2+}$

(Wo-Ca) determines the $CO_2$ consumption associated with the HBEF wollastonite treatment. Our optimistic treatment effect
based on calcium rather than bicarbonate is:
$$\textit{Wo-CO}_{2,Ca}(t) = 2 \times X_{Ca} \times [Ca^{2+}](t) \times \textit{flow}(t), \tag{4}$$
Equations (3) and (4), together with our flux calculations accounting for sparsity of concentration data compared to daily flow
data (Sec. 2.2.5), should help avoid major uncertainties in catchment-scale $CO_2$ consumption calculations: the provenance of
the cations and variations in concentration and discharge (Moon et al., 2014).

### 2.2.5 Flux calculations

To ensure that fluxes from our two watersheds were comparable and to correct for the sparsity of solute measurements
compared to flow measurements, we created rolling annual flow-adjusted fluxes using Method 5 of Littlewood et al. (1998) at
five evenly-spaced points each year:
$$\textbf{Flux} = \text{scale} \times \left[\frac{\sum_{i=1}^{M} C_i Q_i}{\sum_{i=1}^{M} Q_i}\right] \times \left[\frac{\sum_{k=1}^{N} Q_k}{N}\right], \tag{5}$$

where $Q_i$ is the measured instantaneous stream flow, $C_i$ is the concentration for sample $i$, $M$ is the number of streamwater
chemistry samples in the year (usually 12), $Q_k$ is the $k^{th}$ flow measurement, and $N$ is the number of flow measurements. In
our case, daily flow measurements (Campbell, 2015) and ~monthly streamwater samples (Driscoll, 2016b, a) were available.
Therefore, the mean concentration for the preceding twelve months is multiplied by the mean flow for the same period, suitably
scaled to get the total annual flux. Without sub-daily timestamps for the longitudinal streamwater chemistry data, we used
daily total flows rather than instantaneous flows. Tests suggested that there was little difference between using mean daily
instantaneous flows and the mean daily total flows.

### 2.3 Greenhouse gas balance

The success of any treatment for climate change mitigation is determined by the net greenhouse gas ($CO_2$ equivalent) fluxes
prior to and following treatment, at the treatment site and downstream. In addition to increased $CO_2$ consumption, desireable
outcomes for a treatment include increased ecosystem carbon storage in biomass and soils, and decreases in ecosystem,
downstream and logistical greenhouse gas emissions.

### 2.3.1 Greenhouse gas budget for the wollastonite treatment

At the HBEF, we have measured the $CO_2$ consumption due to the wollastonite treatment in two different ways and these
determine our range of values to be incorporated in our GHG budget. Several other treatment effects can be estimated relative
to the reference watershed, but some aspects of the total GHG balance are missing. For example, we have measurements of
soil respiration (root+heterotrophic) and dissolved organic carbon (DOC) export in streamwater, but we lack measurements of
canopy respiration from leaves and stems, and export of particulate organic carbon in streamwater.   Our partial greenhouse
gas budget for the HBEF wollastonite treatment will therefore be given by

$\Delta GHG = \Delta wood + \Delta SRESP + \Delta CH4 + \Delta N2O + \Delta CONS + \Delta NO3N2O + \Delta DOC + LOGPEN$,                                      (6)

where our partial GHG treatment effect ($\Delta GHG$) is the sum of greenhouse gas sink and source responses. Measured sinks for
the wollastonite experiment include biomass in wood ($\Delta wood$), $CO_2$ consumption ($\Delta CONS$), and a soil sink for methane
($\Delta CH4$).  Sources include $N_2O$ emissions both from soil ($\Delta N2O$) and exported nitrate ($\Delta NO3N2O$), and $CO_2$ emissions from
soil respiration ($\Delta SRESP$), exported dissolved organic carbon ($\Delta DOC$), and logistical operations ($LOGPEN$).

210         Sink effects are defined as positive if the sink increases and are given by the difference (treated−reference) between the

two watersheds, whereas source effects are defined as positive for reductions in greenhouse gas emissions (reference−treated).
With these definitions, penalties are negative and reduce $\Delta GHG$ in Eq. (6).  Logistical emissions and $CO_2$ consumption due to
weathering of applied wollastonite are zero for the reference watershed, so we expect $LOGPEN$ to be negative and $\Delta CONS$ to
be positive.

215         Wood is a longer-term carbon sink than leaves or twigs so we have chosen to let this represent our biomass increment.

Eq. (6) neglects ecosystem disturbances including fire, and possible carbonate mineral precipitation in soils. There is no
evidence for the latter at the HBEF.

218         We used a range of emissions factors for $N_2O$ to estimate the penalty associated with nitrate export ($\Delta NO3N2O$); low:

0.0017 $kgN_2O$-N $kg^{-1}$ DIN (Hu et al., 2016) and high: 0.0075 $kgN_2O$-N $kg^{-1}$ DIN (De Klein et al., 2006), where DIN is dissolved
inorganic nitrogen dominated by nitrate.  This $N_2O$ was then converted to $CO_{2e}$ ($CO_2$ equivalents in terms of cumulative
radiative forcing) given the 100-year time horizon global warming potential (Pachauri et al., 2014) ($GWP_{100}$) for $N_2O$: 265
$gCO_{2e}$ $g^{-1}$ $N_2O$. Likewise, $\Delta CH4$ was converted to $CO_{2e}$ ($CO_2$ equivalents in terms of cumulative radiative forcing) given
$GWP_{100}$ for $CH_4$: 28 $gCO_{2e}$ $g^{-1}$ $CH_4$.

**2.3.2 Carbon sequestration in wood**

We calculate our treatment effect on wood production as the difference between the treated and reference watershed mean
wood production  (Battles et al., 2014) over two five-year periods. We considered these differences (treated−reference) to be
an estimate of the treatment effect on potentially long-term (decades to centuries) biomass carbon sequestration. Assuming
46.5% of the woody biomass is carbon (Martin et al., 2018), our calculated cumulative additional C sequestration in the treated
watershed over ten years was 20.7 mol C $m^{-2}$ (9.1 t $CO_2$ $ha^{-1}$).   Our optimistic and pessimistic values are derived from the 95%
confidence intervals for the five-year mean values (Battles et al., 2014).

### 2.3.3 Greenhouse gas emissions from soils

Measurements (Groffman, 2016) were taken at four elevations in the treated watershed and at points just west of the reference watershed starting in 2002 (**Fig. 1**). Gas samples were collected from chambers placed on three permanent PVC rings at each of these eight sites (Groffman, 2016). The data were not normally distributed so were analyzed with Kruskal-Wallis tests at the 0.05 significance level; however, tests with one-way ANOVA produced the same overall results. All analyses were done in Matlab R2016a.

Cumulative curves for each of the 24 chambers were generated by matching the dates of the measurements, excluding points which were missing data for any chamber and allowing up to a week's discrepancy between catchments. Nearly all discrepancies were within one day. Assuming diurnal variation was minor compared to seasonal variation, each datum (g C $m^{-2}$ $hour^{-1}$) was multiplied by 24 hours and by 30 days to get gC $m^{-2}$ $month^{-1}$. There was no extrapolation to fill gaps in the dataset; results are internally consistent but not comparable to other datasets. We were particularly interested in the elevation-specific responses, as the different elevations have distinct tree species compositions and below-ground responses to the wollastonite treatment (Fahey et al., 2016).

The HBEF experimental watersheds are divided into 25×25m plots on slope-corrected grids. Vegetation has been surveyed four times since the late 1990s and assigned a zone designation in each plot (Driscoll et al., 2015;Driscoll Jr et al., 2015;Battles et al., 2015b, a) (Fig. **S12**). To estimate the respiration savings over the whole watershed, we added the areas of individual plots which were assigned to our four vegetation types (Low, Mid and High hardwoods, and Spruce-Fir). Because there were seven vegetation types in the datasets, we compared all types with pairwise Kruskal-Wallis tests at the 0.05 significance level using the basal area data for the six dominant tree species. Kruskal-Wallis tests were appropriate because the data, and therefore the differences from the means (residuals), were not normally distributed. These tests suggested that the "extra" vegetation types ("Birch/Fern Glade", and "Poor Hardwoods" at High and Mid elevations) could be combined with Spruce-Fir, High and Mid Hardwoods respectively. Watershed fractions for our combined forest types were 0.155 for SpruceFir, 0.16 for High Hardwoods, 0.415 for Mid Hardwoods, and 0.27 for Low Hardwoods. When creating our composite treatment effects for the entire watershed, we considered a treatment effect to be present only where our statistical analyses suggested significantly different fluxes.

### 2.3.4 Logistical carbon emissions costs

We used the 1999 upstate New York $CO_2$ emission factor for electricity generation from oil (United States Environmental Protection Agency, 1999) (0.9 Mg $CO_2$ $MWh^{-1}$), and rearranged Equation 28 of Stamboliadis (Stamboliadis et al., 2009):

$$e_p = \frac{\left[ e^{\frac{(\ln s/\alpha)}{\mu}} \right]}{3600 \times 1000}, \tag{7}$$

where the specific surface area $s$ (1600 $m^2$ $kg^{-1}$ for our treatment) is related to the specific potential energy $e_p$ of the material (kJ $kg^{-1}$), with theoretical parameters (Stamboliadis et al., 2009) $\alpha$=139 $m^2$ $kJ^{-1}$ and $\mu$=0.469 (dimensionless). We convert this

potential energy to MWh t$^{-1}$ Qz (3600 seconds per hour and 1000 kWh MWh$^{-1}$). The equation was derived for quartz (Qz)
which has hardness 7. Because wollastonite hardness is in the range 5–5.5, this equation may overestimate the energy needed
to grind the wollastonite.
The main energy source in Allerton will have been coal, and the 1999 Illinois emissions factor (United States
Environmental Protection Agency, 1999) is 1.1 Mg CO$_2$ MWh$^{-1}$. The monetary cost is USD0.041 kWh$^{-1}$ for pelletization of
limestone fines and USD0.85 t$^{-1}$ product, so we estimate 20.73 kWh t$^{-1}$ product.
Road transport distances were estimated using Google Maps (1397 km Gouverneur to Allerton, 1757 km Allerton to
Woodstock, 408 km Gouverneur to Woodstock). We used standard emissions ranges (Sims et al., 2014) for Heavy Duty
Vehicles (HDVs) (70–190 gCO$_2$ km$^{-1}$ t rock$^{-1}$) and for short-haul cargo aircraft (1200–2900 gCO$_2$ km$^{-1}$ t$^{-1}$). Calculation details
are given in Table 2. The Matlab script used for these calculations is available on request. Note: t refers to megagrams, not US
short tons.



**Table 2.** Logistical penalty calculations for the Hubbard Brook wollastonite treatment

| Penalty element | Value and calculation with units |
| --- | --- |
| Mass of wollastonite ($CaSiO_3$) shipped to Allerton ($t^a$ Wo) | 109665 lbs or **49.7432073 t Wo** |
| Mass of pellets shipped from Allerton (t pellets) | 112992 lbs or **51.2523091 t pellets** |
| Ratio of pellet mass to Wollastonite mass | **1.0368** = 51.25 t pellets / 49.74 t Wo |
| HDV transport distance (km) | **3154 km** = 1397 km (Gouverneur to Allerton) + 1757 km (Allerton to Woodstock) |
| Transport distance for "local pelletization" calculation (km) | **408 km** (Gouverneur to Woodstock) |
| Optimistic transport emissions (g $CO_2$ $g^{-1}$ Wo applied) | **0.229 g $CO_2$ $g^{-1}$ Wo applied** = 70 g$CO_2$ $km^{-1}$ shipped $t^{-1}$ shipped $\times$ ((1397 km $\times$49.74 t Wo shipped)+(1757 km $\times$ 51.25 t pellets shipped)) / 48.86 $\times 10^6$ g Wo applied |
| Pessimistic transport emissions (g $CO_2$ $g^{-1}$ Wo applied) | **0.620 g $CO_2$ $g^{-1}$ Wo applied** = 190 g$CO_2$ $km^{-1}$ shipped $t^{-1}$ shipped $\times$ ((1397 km $\times$49.74 t Wo shipped)+(1757 km $\times$ 51.25 t pellets shipped)) / 48.86 $\times 10^6$ g Wo applied |
| Mass of pellets deployed by helicopter (t pellets applied) | 110992 lbs or **50.3451243 t pellets applied** |
| Mass of wollastonite deployed by helicopter (t Wo applied) | **48.86 t Wo applied** = 50.345 t pellets applied / 1.03684 |
| Total area treated (ha) | **14.2 ha** = 11.8 ha watershed plus 2.4 ha "destructive area" along the western edge |
| Nominal mean round trip flight distance (km, Woodstock to watershed and back) | **5 km** |
| Number of flights (1 short ton hopper capacity)[b] | **55.5** = 50.345 t pellets / 0.907 t per trip |
| Molar mass of wollastonite $CaSiO_3$ (g Wo $mol^{-1}$ Wo) | **116.17 g Wo $mol^{-1}$ Wo** = 40.08 g Ca $mol^{-1}$ Ca + 28.09 g Si $mol^{-1}$ Si + 3 $\times$ 16 g O $mol^{-1}$ O |
| Molar mass of $CO_2$ (g $CO_2$ $mol^{-1}$ $CO_2$) | **44.01 g $CO_2$ $mol^{-1}$ $CO_2$** = 2 $\times$ 16 g O $mol^{-1}$ O + 12.01 g C $mol^{-1}$ C |
| Optimistic spreading emissions (mol $CO_2$ $ha^{-1}$) | **483.36 mol $CO_2$ $ha^{-1}$** = 1200 g$CO_2$ $km^{-1}$ $t^{-1}$ $\times$ 5 km $\times$ 50.345 t pellets / 44.01 g $CO_2$ $mol^{-1}$ $CO_2$ / 14.2 ha |
| Optimistic spreading emissions (g $CO_2$ $g^{-1}$ Wo) | **0.006 g $CO_2$ $g^{-1}$ Wo** = 1200 g$CO_2$ $km^{-1}$ $t^{-1}$ $\times$ 5 km $\times$ 50.345 t pellets /48.86/$10^6$ g Wo |
| Pessimistic spreading emissions (mol $CO_2$ $ha^{-1}$) | **1168.1 mol $CO_2$ $ha^{-1}$** = 2900 g$CO_2$ $km^{-1}$ $t^{-1}$ $\times$ 5 km $\times$ 50.345 t pellets / 44.01 g $CO_2$ $mol^{-1}$ $CO_2$ / 14.2 ha |
| Pessimistic spreading emissions (g $CO_2$ $g^{-1}$ Wo) | **0.015 g $CO_2$ $g^{-1}$ Wo applied**= 2900 g$CO_2$ $km^{-1}$ $t^{-1}$ $\times$ 5 km $\times$ 50.345 t pellets / 48.86 $\times 10^6$ g Wo applied |


[a]Megagrams or metric tons, not short tons
[b]Number of flights does not explicitly enter into penalty calculations because the emissions for shorthaul aircraft are multiplied
by the 5km round trip distance and the entire mass transported, rather than the mass transported during one round trip (one
short ton).


## 3 Results

### 3.1 Wollastonite treatment increased streamwater CO₂ export

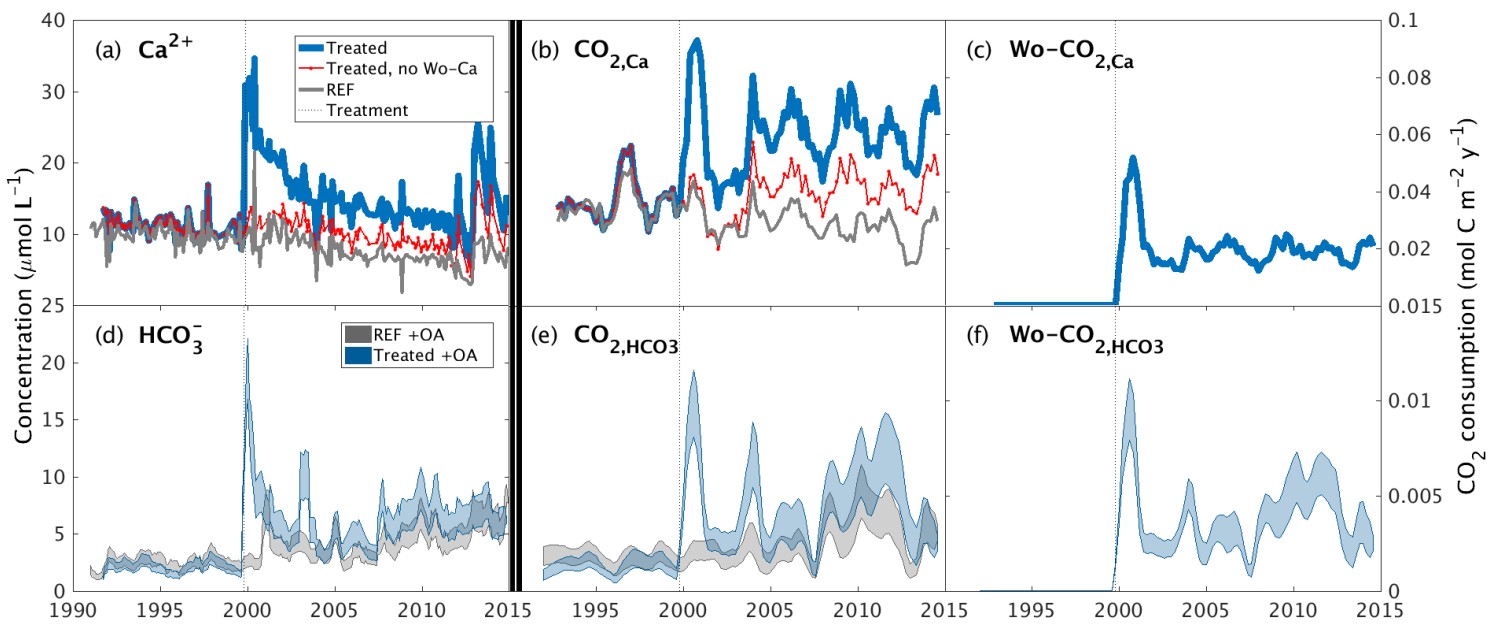

**Figure 2: Inorganic CO₂ capture at the Hubbard Brook Experimental Forest.** (a) Observed calcium and (b) calcium export in the reference (grey) and treated (blue) watersheds along with the contribution from sources other than wollastonite (red) and the time of treatment (vertical dotted line). (c) Calculated CO₂ consumption due to the treatment (*Wo-CO$_{2,Ca}$*, Eq. 9). (d) Modelled streamwater bicarbonate, (e) CO₂ consumption (*CO$_{2,HCO3}$*, Eq. 5), and (f) CO₂ consumption due to the treatment (*Wo-CO$_{2,HCO3}$*, Eq. 7), colours as for calcium. Simulations (d–f) account for the presence of organic acids (+OA). All calcium export (b) and CO₂ consumption curves (c,e,f) were calculated with flow-normalised concentrations and corrected for sparsity of samples (Methods).

We first consider the time-series of streamwater changes in Ca$^{2+}$ concentrations in the treated ([Ca]$_{Treated}$) and reference ([Ca]$_{Ref}$) watersheds. Immediately after treatment, [Ca]$_{Treated}$ increased from <30 μmol L$^{-1}$ to ~60 μmol L$^{-1}$, and then slowly declined over the next decade, remaining persistently above [Ca]$_{Ref}$ for 15 years (**Fig. 2a**). The initial post-treatment peak represents dissolution of wollastonite within the stream (Peters et al., 2004) and release of calcium from hyporheic exchange during the first few years (Shao et al., 2016;Nezat et al., 2010). Retention of Ca$^{2+}$ ions liberated by wollastonite dissolution (Wo-Ca) in the watershed soils (Nezat et al., 2010) and sequestration into tree biomass (Balogh-Brunstad et al., 2008;Nezat et al., 2010) delayed appearance in streamwater for three years (Shao et al., 2016;Nezat et al., 2010). Subsequently, [Ca]$_{Treated}$ remained approximately double [Ca]$_{Ref}$, with a ~30% contribution from non-wollastonite Ca$^{2+}$ until 2012. Towards the end of the time-series, increased seasonal NO$_3^-$ export in the treated watershed between 2012 and 2014 (Rosi-Marshall et al., 2016) led to Wo-Ca displacing non-Wo-Ca from the soil exchanger.

We derived the annual export of $Ca^{2+}$ from the treated and reference watersheds as the product of mean annual flow-
adjusted $Ca^{2+}$ streamwater concentrations and annual flow (**Fig. 2b**) (Methods). After accounting for variations in flow,
increased streamwater $Ca^{2+}$ concentrations in the treated watershed are translated into a 2-fold increase in total $Ca^{2+}$ export
relative to the reference watershed that was maintained for 15 years until 2015 through this analysis period. Overall, the
wollastonite treatment resulted in a sharp spike in calculated $CO_2$ consumption (***Wo-CO_{2,Ca}***) that decreased but remained
elevated as a result of the treatment (**Fig. 2c**).
Temporal patterns in modelled streamwater bicarbonate concentration in both treated and reference watersheds (**Fig.**
**2d**), and the corresponding total annual $CO_2$ consumption (***CO_{2,HCO3}***) (**Fig. 2e**) and $CO_2$ consumption resulting from treatment
(***Wo-CO_{2,HCO3}***) (**Fig. 2f**), largely mirror changes in streamwater $Ca^{2+}$ concentrations but are modified by the supply and loss of
anions. Calculated flow-adjusted $CO_2$ consumption (**Fig. 2e**) peaked 2–3 years post-treatment with a broader peak in $CO_2$
consumption evident in 2007–2012 corresponding to declining legacy effects of acid rain until transient $NO_3^-$ peaks appeared
2012–2015. ***Wo-CO_{2,HCO3}*** shows a pattern that mirrors ***Wo-CO_{2,Ca}*** but is generally 5 times lower (**Fig. 2c,f**).
**3.2 Sulphuric, nitric and organic acids reduce CDR**
We next undertook sensitivity analyses to investigate the effects of acid deposition, increased $NO_3^-$ and organic acid export
from the treated watershed on bicarbonate concentrations and resulting $CO_2$ consumption (**Fig. S5**). In a 'Low $SO_4$' scenario
(**Fig. S5a–c**), we sought to understand the effects of acid deposition by replacing the mean monthly time-series of streamwater
and rainwater $SO_4^{2-}$ for the treated watershed with a new time-series (purple curve, **Fig. S5a**) created by repeating the post-
2010 datasets, which reflect diminished acid deposition following emission controls from the US Clean Air Act (Likens and
Bailey, 2014). Removing acid rain effects in this manner dramatically increased the calculated bicarbonate concentrations and
total annual $CO_2$ consumption (***CO_{2,HCO3}***), increasing the initial spikes resulting from the wollastonite treatment in both by at
least four-fold (purple curves, **Fig. S5 b,c**). An additional legacy of acidification in North American forests (Harrison et al.,
1989) is $SO_4^{2-}$ retention on soil clay mineral Fe and Al oxides (Fuller et al., 1987), which were subsequently released by
increased soil pH following wollastonite weathering (Shao et al., 2016;Fakhraei et al., 2016). To assess the effect of this
legacy $SO_4^{2-}$, we ran simulations for the treated watershed substituting the lower streamwater $SO_4^{2-}$ concentrations from the
reference watershed (T REF, green curves, **Fig. S5b,c**). Results suggest that legacy $SO_4^{2-}$ accounts for over half of the total
acid deposition effect on increased $[HCO_3^-]$ and $CO_2$ consumption in the simulations.
In the 'Ref $NO_3$' scenario (**Fig. S5 d–f**), seasonal spikes in streamwater export of $NO_3^-$ recorded from the treated
watershed between 2012 and 2015 were removed by substituting the reference watershed streamwater $NO_3^-$ concentration
measurements lacking these spikes. This manipulation markedly increased modelled bicarbonate (**Fig. S5e**) and mean annual
$CO_2$ consumption (**Fig. S5f**). To quantify the effects of organic acids on bicarbonate production in the treated watershed, we
ran "+OA" and "-OA" simulations, i.e., with and without accounting for organic acids, respectively (**Fig. S5 g–i**). Results
showed that removing OA from our simulations also increased modelled streamwater bicarbonate concentration (**Fig. S5h**),
and resulting $CO_2$ consumption (**Fig. S5i**), in the treated watershed.

## 3.3 Effects of increasing wollastonite treatment

Because the HBEF application rate (3.44 t ha$^{-1}$) is smaller than the 10–50 t ha$^{-1}$ suggested for ERW strategies (Strefler et al., 2018;Beerling et al., 2018), we simulated the possible effects of a ten-fold increase in the streamwater Ca$^{2+}$ concentrations on bicarbonate production (**Fig. S6a**) and $CO_2$ consumption (**Fig. S6b**). In this initial assessment, we assume streamwater responses are directly proportional to wollastonite application rate, i.e., 34.4 t ha$^{-1}$, and that all other variables remained unchanged. Results show that after 15 years, cumulative ***Wo-CO$_{2,HCO3}$*** is 73% of ***Wo-CO$_{2,Ca}$*** (**Fig. S6c**), as opposed to less than 20% for the actual rate of 3.44 t ha$^{-1}$ (**Table 3**). These results suggest that at higher application rates of wollastonite, the details of the $CO_2$ consumption calculations become less important.

**Table 3.** Cumulative fluxes from treatment date calculated with streamwater partial pressure of $CO_2$ (gas) = 3.63 × atmospheric $CO_2$ partial pressure measured at Mauna Loa (Tans and Keeling, 2017) (see Methods). DIC = dissolved inorganic carbon. Scenarios are defined in the main text.

| Cumulative fluxes 1 year post-treatment date (19 October 2000) | | | | | | | | | |
|---|---|---|---|---|---|---|---|---|---|
| **Watershed** | **Scenario** | **Org. acids** | ***CO$_{2,ions}$*** **(Eq. 2)** | ***Wo-CO$_{2,Ca}$*** **(Eq. 4)** | **DIC** | **HCO$_3$** | ***CO$_{2,HCO3}$*** **(Eq. 1)** | ***Wo-CO$_{2,HCO3}$*** **(Eq. 3)** |
| | | | mol C m$^{-2}$ | | | | | | |
| REF (6) | baseline | +OA | –0.003 | 0 | 0.084 | 0.002 | 0.002 | 0 |
| Treated (1) | baseline | +OA | 0.047 | 0.052 | 0.086 | 0.011 | 0.011 | 0.011 |
| Treated (1) | baseline | –OA | 0.047 | 0.052 | 0.094 | 0.019 | 0.019 | 0.018 |
| Treated (1) | Low SO4 | +OA | 0.083 | 0.052 | 0.117 | 0.043 | 0.042 | 0.039 |
| Treated (1) | REF NO3 | +OA | 0.047 | 0.052 | 0.105 | 0.030 | 0.030 | 0.029 |
| Treated (1) | WoX10 | +OA | 0.513 | 0.534 | 0.533 | 0.457 | 0.457 | 0.457 |
| Cumulative fluxes 15 years post-treatment (20 November 2014) | | | | | | | | | |
| **Watershed** | **Scenario** | **Org. acids** | ***CO$_{2,ions}$*** **(Eq. 2)** | ***Wo-CO$_{2,Ca}$*** **(Eq. 4)** | **DIC** | **HCO$_3$** | ***CO$_{2,HCO3}$*** **(Eq. 1)** | ***Wo-CO$_{2,HCO3}$*** **(Eq. 3)** |
| | | | mol C m$^{-2}$ | | | | | | |
| REF (6) | baseline | +OA | –0.274 | 0 | 1.307 | 0.052 | 0.036 | 0 |
| Treated (1) | baseline | +OA | –0.044 | 0.294 | 1.299 | 0.083 | 0.064 | 0.057 |
| Treated (1) | baseline | –OA | –0.044 | 0.294 | 1.414 | 0.198 | 0.179 | 0.145 |
| Treated (1) | Low SO4 | +OA | 0.269 | 0.294 | 1.523 | 0.307 | 0.270 | 0.179 |
| Treated (1) | REF NO3 | +OA | –0.044 | 0.294 | 1.410 | 0.194 | 0.175 | 0.127 |
| Treated (1) | WoX10 | +OA | 2.600 | 3.275 | 3.626 | 2.406 | 2.387 | 2.380 |


### 3.4 Amplification of organic carbon sequestration by wollastonite treatment

In reversing long-term $Ca^{2+}$ depletion of soils, the silicate rock treatment significantly increased forest growth and wood production between 2–12 years post-treatment relative to the reference watershed (Battles et al., 2014). This forest response increased total carbon sequestration by 20.7 mol C m$^{-2}$ or 9.1 t $CO_2$ ha$^{-1}$ during those ten years as a result of the treatment (Methods).

Changes in greenhouse gas (GHG) emissions from soils represent a further route to affecting the climate mitigation potential of the wollastonite treatment. Despite a rapid increase of one pH unit in the upper organic soil horizon (Oie), soil respiration $CO_2$ fluxes showed no significant difference between watersheds during the first three years after treatment (Groffman et al., 2006). However, our analysis of newly available longer-term datasets indicates that the treatment significantly reduced soil respiration in the high elevation hardwood zone (~660–845m a.s.l.) ($\chi^2(1,270)=17.2$, $P < 0.001$), possibly due to reduced fine-root biomass (Fahey et al., 2016) rather than changes in microbial activity (Groffman et al., 2006). No significant effects on soil respiration were detected in any of the other HBEF vegetation zones (**Fig. 3**). The wollastonite treatment increased the soil sink strength for $CH_4$ ($\chi^2(1,266)=30.8$, $P < 0.001$) in the low-elevation hardwood zone (482–565m a.s.l.), while it decreased in the high elevation zone ($\chi^2(1,268)=22.3$, $P < 0.001$) (SI Appendix, Fig. S8). There were no significant treatment effects on soil $N_2O$ fluxes in any vegetation zone (SI Appendix).

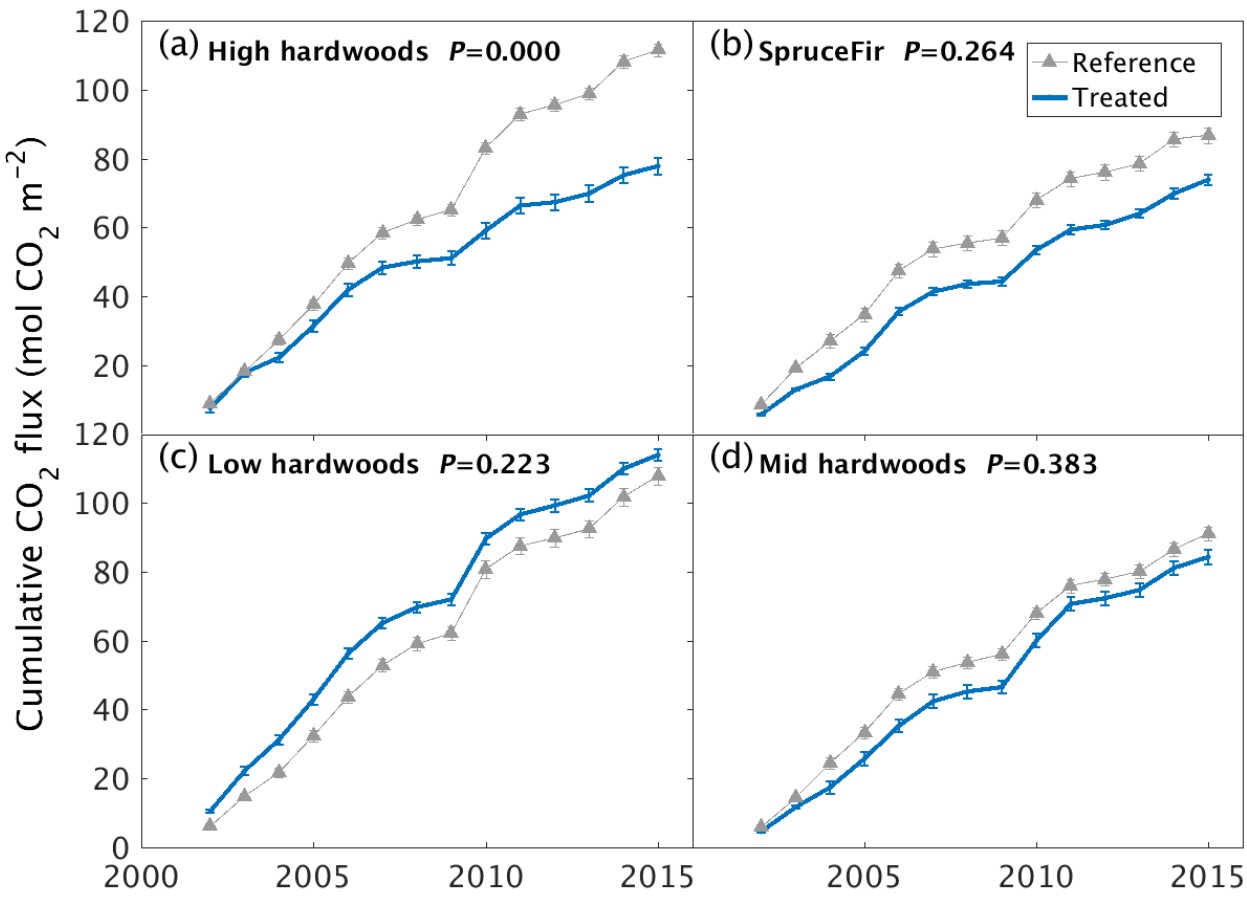

370

**Figure 3: Long-term soil respiration responses to wollastonite treatment at Hubbard Brook Experimental Forest.** Cumulative soil $CO_2$ respiration responses of treated and untreated (a) high elevation hardwoods, (b) high elevation conifers, (c) low elevation hardwoods or (d) mid-elevation hardwoods. Plots show cumulative means ± 1 SE for three chamber measurements at each site and time. Reference data were collected from untreated forests immediately adjacent to the western edge of our reference catchment. *P*-values from Kruskal-Wallis tests comparing treated and reference raw data (SI Appendix) are shown.

376

### 3.5 Logistical $CO_2$ emissions and net CDR

We next considered carbon emissions (penalties) for logistical operations involved in mining, grinding, transporting and applying the wollastonite (**Fig. 4, Table 2**). In the HBEF experiment, wollastonite was mined and milled on site near Gouverneur, New York. We used $CO_2$ emissions factors for electricity generation in upstate New York (United States Environmental Protection Agency, 1999) to estimate the maximum $CO_2$ penalty for mining and grinding to the mean particle

size 16 µm diameter (Methods). However, local hydropower (Energy Information Administration, 1997) and regional nuclear
power suggest these costs could have been zero. This would represent a substantial carbon saving for the overall ERW process
relative to prior expectation of ERW studies in which grinding $CO_2$ emissions account for up to 30% reduction in ERW-CDR
efficiency (Renforth, 2012;Moosdorf et al., 2014).

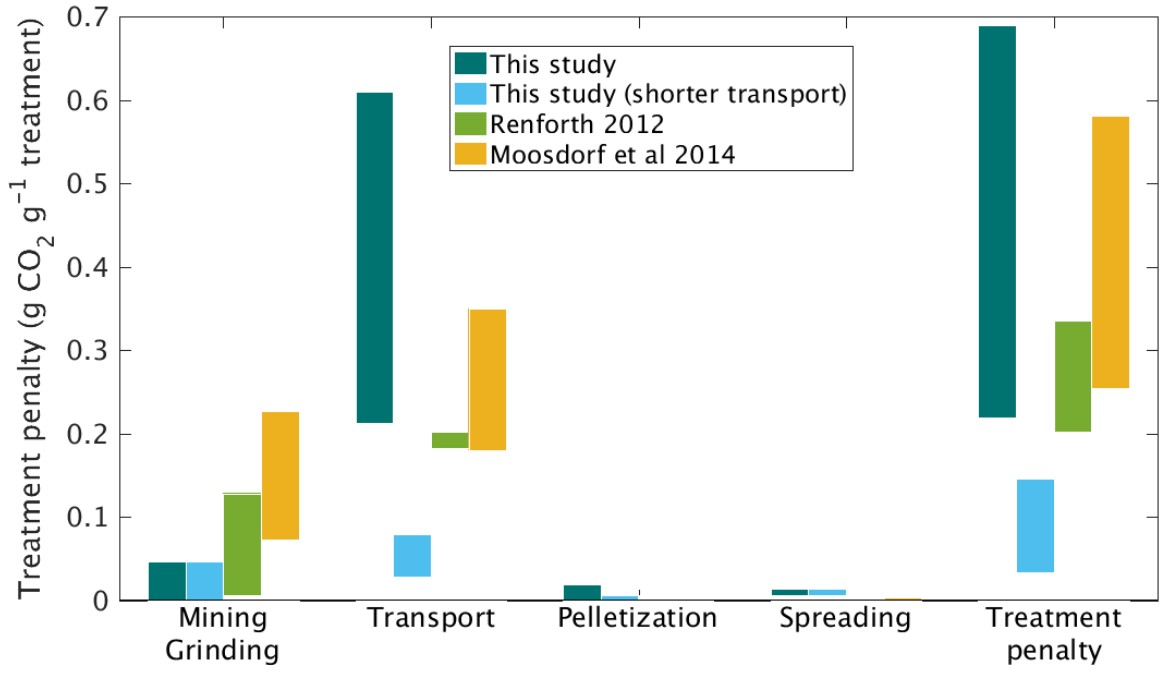


**Figure 4: Carbon penalties for the wollastonite treatment.** Carbon penalties for logistic elements of the treatment are compared with
literature estimates for large-scale rollout of enhanced rock weathering for the HBEF treatment (3.44 t ha[-1]), with and without long-distance
transport for pelletization.

391       In the HBEF experiment, the milled wollastonite was transported by highway to Allerton, Illinois, for pelletization

and then returned to the staging area near Woodstock, New Hampshire (round trip >3150 km). Transportation $CO_2$ emissions
were 0.22–0.61 t $CO_2$ t Wo[-1]. Given coal power in central Illinois, we estimate pelletization emitted up to 0.02 t $CO_2$ t Wo[-1]
(Methods). Application at Hubbard Brook occurred via 55 ~5-km helicopter flights, which gives a further $CO_2$ cost of 0.01–
0.15 t $CO_2$ t Wo[-1]. In total, these logistical operations emitted 0.23–0.69 t $CO_2$ t Wo[-1], or 0.8–2.4 t $CO_2$ ha[-1] for the 11.8 ha of
the HBEF treated watershed (**Table 4**). However, local pelletization could have reduced heavy duty vehicle (HDV) transport
distance to ~400 km and lowered total $CO_2$ emitted during logistical operations to 0.04–0.15 t $CO_2$ t Wo[-1]. At other forested
sites, where wind-drift of material is not critical, pelletization may not be necessary.

**Table 4.** Measured elements of the treatment effect on the greenhouse gas budget for the Hubbard Brook Experimental Forest
wollastonite experiment.

| Equation 14 | Greenhouse gas sinks[a] and emissions[a] (t $CO_{2e}$ ha$^{-2}$) | Pessimistic | Optimistic |
|---|---|---|---|
| **Ecosystem responses[b]** | | | |
| *Δwood* | Wood production sink increased over ten years[c] | 8.946 | 9.542 |
| *ΔSRESP* | Soil respiratory $CO_2$ emissions have reduced[a] since 2002 | 2.213 | 2.646 |
| *ΔCH4* | Soil methane sink has increased since 2002 | 0.015 | 0.029 |
| *ΔN2O* | Soil $N_2O$ emissions since 2002 (no significant difference) | 0 | 0 |
| | **Net ecosystem response at the treatment site through 2014** | **11.174** | **12.218** |
| **Downstream sequestration and emissions responses** | | | |
| *ΔCONS* | $CO_2$ consumption sink through 2014 (*Wo-CO$_{2,HCO3}$* and *Wo-CO$_{2,Ca}$*) | 0.025 | 0.129 |
| *ΔNO3N2O* | Downstream $N_2O$ emissions[d] from treatment date through 2014 | –0.071 | –0.016 |
| *ΔDOC* | DOC export emissions[d,e] from treatment date through 2014 | –0.203 | 0 |
| | **Net downstream balance through 2014** | **–0.228** | **-0.129** |
| **Logistics:** | | | |
| | Mining/Grinding given hydro or nuclear/petroleum power | –0.162 | 0 |
| | Helicopter (~55 5-km flights) | –0.051 | –0.021 |
| | HDV transport (New York to Illinois to New Hampshire) | –2.135 | –0.787 |
| | Pelletization (in Illinois, coal power) | –0.068 | 0 |
| *LOGPEN* | **Total logistical emissions** | **–2.416** | **–0.808** |
| *ΔGHG* | **Partial treatment effect on greenhouse gas balance** | **8.509** | **11.523** |

[a]Defined as the difference between watersheds: treated–reference for sinks and reference–treated for emissions
[b]Some possible treatment responses such as canopy respiration and particulate organic carbon export are unknown.
[c]After Battles et al. (2014). We have not attempted to extrapolate these results.
[d]ΔDOC and ΔNO3N2O are penalties because these lead to $CO_2$ and $N_2O$ emissions downstream.
[e]The "optimistic" value for DOC assumes complete burial and undesireable low oxygen conditions in downstream waters.


409       These carbon emission penalties must be subtracted from watershed carbon removal to calculate net CDR for the

wollastonite treatment at HBEF (**Fig. 5**; **Table 4**). Compared in this way, we find increased wood production over ten years
(Battles et al., 2014) repays the total logistical $CO_2$ costs 4–12 times over. The components (**Fig. 5; Table 4)** comprise 8.5–
11.5 t$CO_2$ ha$^{-1}$ of the total GHG budget associated with the wollastonite treatment (Methods). These figures would increase
to 10.4–12.2 t$CO_2$ ha$^{-1}$ if the wollastonite had been pelletized anywhere along the route from Gouverneur to New Hampshire.

414       Wollastonite treatment effects on streamwater chemistry play a minor role in the greenhouse gas budget (**Fig. 5;**
**Table 4**). For our hypothetical ten-fold higher treatment (34.4 t ha$^{-1}$), $CO_2$ consumption calculated by assumed calcium release
is ~10 times higher, but carbon emission penalties scale with increased rock mass. Assuming pelletization near the mine to
reduce transport costs, the total logistical penalty would be 1.2–5.1 t$CO_2$ ha$^{-1}$. In total, net CDR would be 6.8–12.4 t$CO_2$ ha$^{-1}$
for the the ten-fold larger treatment if none of the other GHG fluxes changed. We have not attempted to extrapolate other forest
biomass and soil GHG fluxes or streamwater DOC and $NO_3^-$ responses.

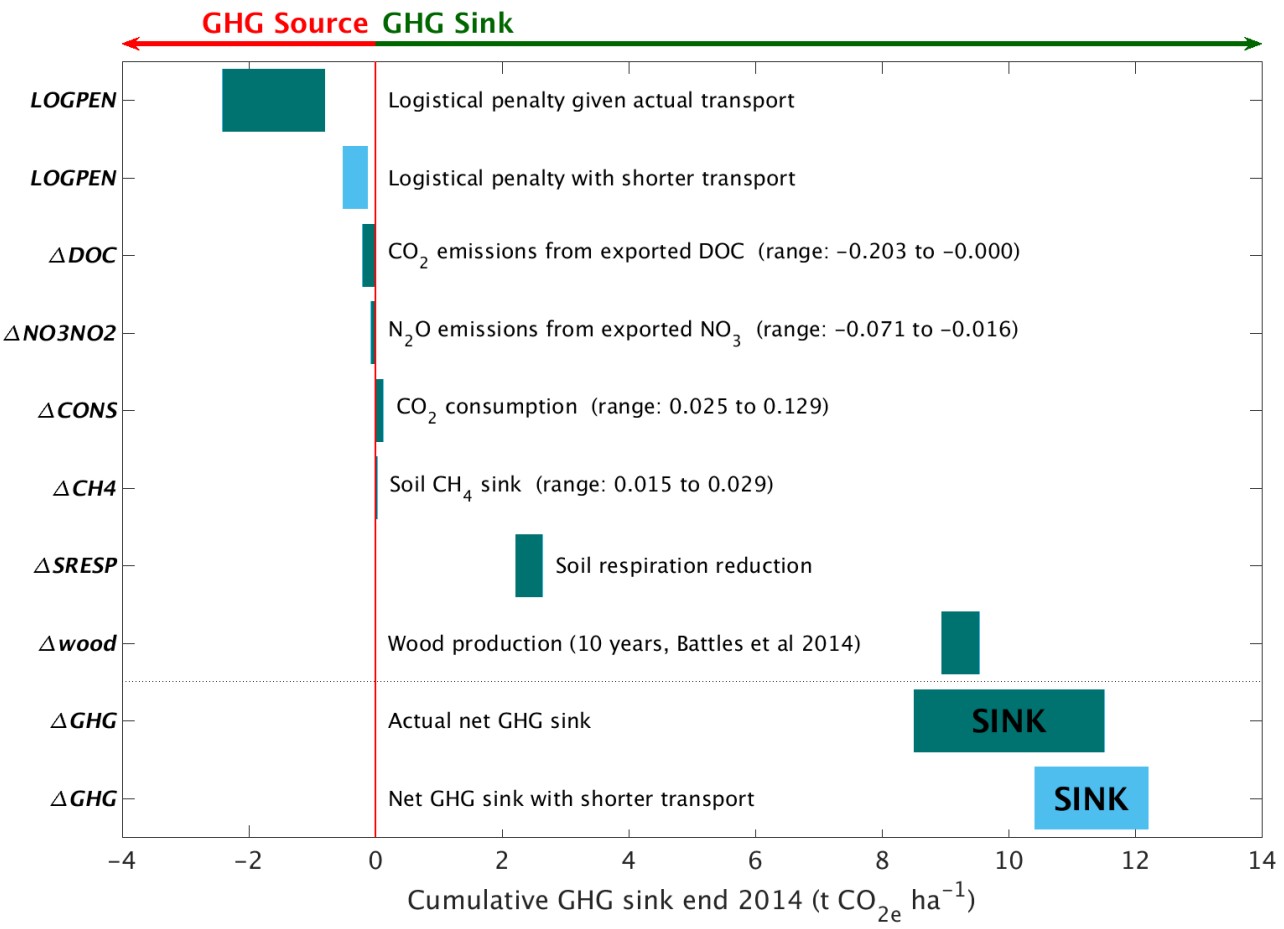

**Figure 5: Carbon responses for the wollastonite treatment.** Elements of the greenhouse gas balance associated with the wollastonite
treatment (**Table 4**). The $CO_2$ consumption range is given by ***Wo-CO$_{2,HCO3}$*** calculated by Eq. (3) and ***Wo-CO$_{2,Ca}$*** calculated by Eq. (4), time-
integrated from the application date through 2014. Nitrate export in streamwater leading to $N_2O$ greenhouse gas emissions downstream and
a small increase in the soil $CH_4$ sink have been converted to $CO_2$-equivalents (Methods). Exported DOC is assumed to be respired
downstream.

### 3.6 Potential for deployment at larger scales

The HBEF forests are representative of a major area of eastern North America receiving acid deposition since the 1950s (Likens and Bailey, 2014) which may be suitable for remediation and carbon capture via ERW treatment with a silicate rock or mineral. For example, the Appalachian and Laurentian-Acadian Northern Hardwood Forests (NHWF) covering a combined area of 0.137 Mkm$^2$ in the United States (Ferree and Anderson, 2013) have the same dominant hardwood trees as the HBEF experimental watersheds (*Fagus grandifolia, Betula alleghniensis* and *Acer saccharum*). Acid deposition exceeded "critical loads" likely to harm ecosystems in almost 9000 ha of New Hampshire's *Acer saccharum* stands (NHAs) (Schaberg et al., 2010). These forests might be expected to respond similarly to a wollastonite treatment. The acid-sensitive trees *Acer saccharum* and *Picea rubens* are also widely distributed along the high elevation acid sensitive regions of the Appalachian Mountains which have already been impacted by acid deposition (Lawrence et al., 2015). We define this as a 40-km corridor along the Appalachian Mountains comprising 0.14 Mkm$^2$ and overlapping with the High Allegheny Plateau Ecoregion (HAL) where *Acer saccharum* is declining above ~550 m a.s.l. (Bailey et al., 2004) (0.07 Mkm$^2$).

We examined the potential $CO_2$ consumption for a range of wollastonite application rates encompassing those suggested for ERW strategies (Strefler et al., 2018;Beerling et al., 2018) (**Fig. 6**). In this analysis, we adjusted mean (2003–2012) Wo-$CO_{2,ca}$ for the actual 3.44 t ha$^{-1}$ treatment (~0.2 mol C m$^{-2}$ yr$^{-1}$) proportionally for 10–50 t ha$^{-1}$ treatments. We assume logistical carbon penalties are minimised and balanced by forest biomass carbon sequestration responses to treatment. This analysis suggests net CDR potential of 0.3–1.7 Mt $CO_2$ yr$^{-1}$ along the Appalachian corridor, which is 2–12% of New Hampshire state emissions (13.8 Mt $CO_2$) in 2016 (Energy Information Administration, 2019). However, world wollastonite reserves (Curry, 2019) ($\geq$0.1 Pg) are insufficient to treat large areas of eastern North America at rates of 10–50 t ha$^{-1}$, highlighting the requirement for alternative sustainable sources of silicate materials.

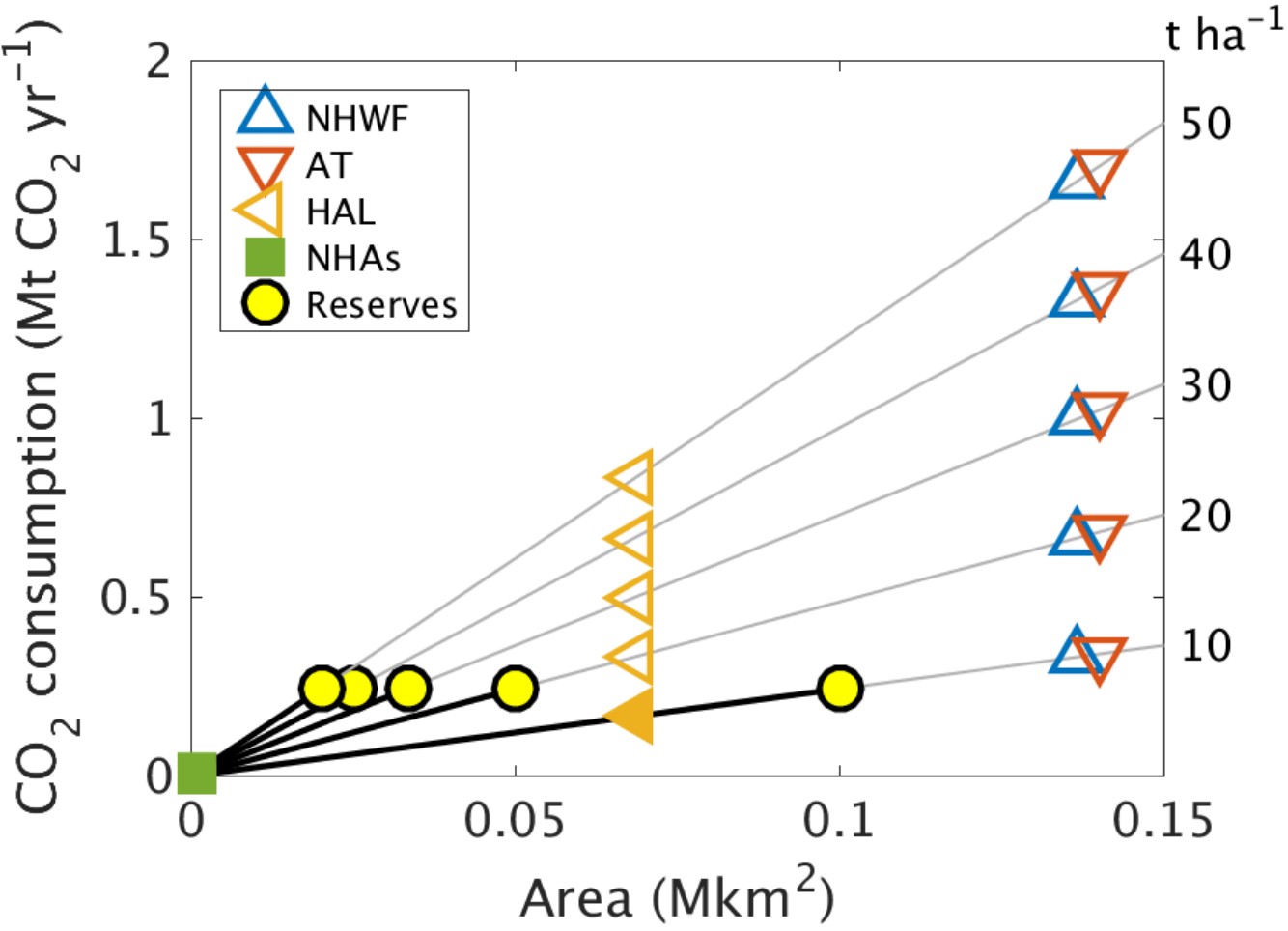

452

**Figure 6: Projected CO₂ consumption following higher-dosage treatments.** We considered the possibility of higher-dosage silicate treatments on other northeastern United States higher-altitude forests affected by acid rain, such as *Acer saccharum* forests in New Hampshire (NHAs), the High Allegheny Plateau Ecoregion (HAL), the Appalachian trail corridor (AT), or Northern Hardwood forests (NHWF) dominated by the same tree species as at Hubbard Brook. Because the world's wollastonite reserves (yellow disks) are insufficient to treat these areas, other calcium-rich silicate minerals would be required. $CO_2$ consumption due to higher dosage (t ha$^{-1}$) is estimated as: (mean observed $CO_{2,Ca}$ between 2004 and 2012) × area × dosage / 3.44 t ha$^{-1}$.


## 4 Discussion

Our analyses of wollastonite application at the HBEF provide a unique long-term (15 year) perspective on the whole watershed
carbon cycle responses and net CDR by accounting for the associated $CO_2$ costs of logistical operations. By 2015, net CDR
amounted to 8.5–11.5 t $CO_2$ ha$^{-1}$ at a low rate of wollastonite application, with increased carbon sequestration into forest
biomass playing the dominant role. We estimate that if the HBEF application rates were increased ten-fold, net CDR would

increase by 8%, assuming 400-km transport distances and no change in forest responses. Amplification of organic carbon capture may therefore represent a major CDR benefit of ERW when applied to forested lands affected by acid rain. Forest management practices, disturbance regimes and the ultimate fate of any harvested wood are also important in determining the storage lifetime of the sequestered carbon. Our results highlight the need to carefully monitor the net carbon balance of forested ecosystems in response to a silicate treatment, including wood and canopy respiration (Fahey et al., 2005) (Methods). This challenging goal might best be achieved with fully instrumented eddy covariance plots, although the HBEF topography is not well suited for this approach (Fahey et al., 2005).

Inorganic $CO_2$ consumption calculated based on streamwater bicarbonate fluxes approximately doubled in the treated watershed relative to the reference watershed 15 years post-treatment (0.028 and 0.016 $tCO_2$ ha$^{-1}$, respectively) (**Table 3**). The presence of $SO_4^{2-}$, $NO_3^-$ and organic acid anions lowered the efficiency of $CO_2$ consumption by alkalinity generation, with acid deposition having the single largest calculated effect (**Table 3**). The cause of increased $NO_3^-$ export from the treated watershed is not as yet understood (Rosi-Marshall et al., 2016). If it proves a general feature of terrestrial ecosystem responses to silicate mineral treatment, this could affect the efficiency of carbon capture via bicarbonate export. Overall, we suggest that continued recovery of eastern North American and European forests and soils from acid deposition creates conditions beneficial to watershed health, carbonic acid-driven weathering and inorganic carbon export following application of crushed silicate minerals.

In Asia, acid rain is an ongoing problem with an estimated 28% of Chinese land area (~2.7 Mkm$^2$) receiving potentially damaging S deposition in 2005 (Zhao et al., 2009), and critical loads were exceeded in ~0.36 Mkm$^2$ of the European Economic Area (EEA) in 1999 (Larssen et al., 2003), approximately double the affected area of US Northern Hardwood Forests (**Fig. 6**). **Fig. 6** suggests that a single 30t Wo ha$^{-1}$ treatment over 0.14 Mkm$^2$ (Appalachian Trail corridor) could, in principle, sequester ~1 $MtCO_2$ y$^{-1}$ or 15 $MtCO_2$ over 15 years via wollastonite-derived Ca export in streamwater alone. Adding the Chinese and European acidified areas could potentially sequester 0.34 $GtCO_2$, approximately 0.2–0.7% of the ~50–150 Gt CDR required by 2050 to avoid warming in excess of 1.5° (Rogelj et al., 2018). Inclusion of biomass and soil responses increases CDR contributions from ERW on acidified forests, but these will still be modest. Assuming no further forest responses beyond the 15-year HBEF timeframe, we report a GHG balance of ~10 $tCO_{2e}$ ha$^{-1}$. This translates to 1 $GtCO_{2e}$ Mkm$^{-2}$ suggesting 3.2 $GtCO_{2e}$ over 15 years for the Appalachian Trail, the EEA and China combined, or 2–6% of global required CDR as described above.

It is uncertain whether other acidified forest ecosystems would respond similarly to the HBEF *Acer saccharum* forests in New Hampshire. Many Chinese soils (Duan et al., 2016), as well as old deep soils in areas such as the Virginian Blue Ridge Mountains and the German Harz and Fetchel Mountains (Garmo et al., 2014), have high $SO_4^{2-}$ sorption capacity. These soils may retain substantially more $SO_4^{2-}$ than the HBEF soils, with potential for prolonged $SO_4^{2-}$ flushing following ERW treatment and lower bicarbonate production. Liming studies suggest a range of other effects, some of which may also occur with silicate treatments. Liming increases nitrate export, migration of heavy metals and acidity to deeper soil, and fine root production in topsoils leading to frost damage (Huettl and Zoettl, 1993).

Many forests have been limed with carbonate minerals such as calcite and dolomite to mitigate acidification in the past.
Dolomite has also helped reverse Mg deficiency in conifers (Huettl and Zoettl, 1993). Liming generally improves water
quality, although it also forms mixing zones with high-molecular-weight Al complexes toxic to fish (Teien et al., 2006). With
silicate treatments, nontoxic hydroxyaluminosilicates form instead (Teien et al., 2006). Unfortunately, carbonates are
contraindicated for CDR on acid soils because they can be a net source of $CO_2$ in the presence of strong acids (Hamilton et al.,
2007). Treatments of European and North American acidified forests with calcite (1–18 t ha$^{-1}$ $CaCO_3$) or dolomite (2–8.7 t
ha$^{-1}$ $CaMg(CO_3)_2$) have, in general, resulted in increased DOC export and soil respiration without increasing tree growth,
regardless of forest composition (Lundström et al., 2003). As calcite and dolomite are 44% and 48% $CO_2$ by weight, these
treatments will have released 0.44–7.9 and 0.96–4.54 t $CO_2$ ha$^{-1}$ respectively when fully dissolved, although dissolution may
be slow. Over six years following a 2.9 t dolomite ha$^{-1}$ treatment (90% 0.2–2.0 mm grains) in a Norwegian coniferous
watershed equating to 1.36 t $CO_2$ ha$^{-1}$, less than 1% of the dolomite dissolved (Hindar et al., 2003). We estimate that $CO_2$
consumption corrected for $CO_2$ release and as measured with dolomite-derived Ca and Mg in streamwater (Dol-$CO_{2,Ca+Mg}$)
averaged 0.02 mol $CO_2$ m$^{-2}$ yr$^{-1}$. $CO_2$ release from carbonate minerals equals Ca and Mg release on a molar basis, so 0.02 mol
Dol-$CO_2$ m$^{-2}$ yr$^{-1}$ was also either exported in streamwater or lost to the atmosphere. This experiment may have a negative
greenhouse-gas balance depending on logistical penalties and soil respiration, as there was no significant treatment effect on
tree growth or vitality (Hindar et al., 2003). Ca-sensitive *Acer saccharum* is present at Woods Lake in New York State, yet
tree biomass decreased with no significant differences relative to reference catchments during the 20 years following a 6.89 t
Mg-calcite ha$^{-1}$ application (Melvin et al., 2013), equivalent to 3.07 t $CO_2$ ha$^{-1}$ given 8% Mg content of the pellets. In contrast
to our study and other liming studies, root biomass and soil carbon stocks increased in response to this treatment, although soil
respiration was reduced (Melvin et al., 2013). *Acer saccharum* basal area and crown vigour increased over 23 years in response
to 22.4 t dolomitic limestone ha$^{-1}$ (equivalent to 10.0 t $CO_2$ ha$^{-1}$) on the Allegheny Plateau, although basal area and survival of
another dominant canopy species, *Prunus serotina*, was reduced (Long et al., 2011). Clearly, forest responses to mineral
treatments are species- and site-specific.
Although the HBEF experiment used wollastonite, this is not a target mineral for ERW, both because of its limited
reserves (Curry, 2019) and high monetary costs (Schlesinger and Amundson, 2018). Recent all-inclusive guide prices of ~700
USD Mg$^{-1}$ for helicopter deployment of pelletized lime along the Appalachian Mountain corridor are comparable to the price
of 694 USD Mg$^{-1}$ for unpelletized 10-µm wollastonite in 2000 (Virta, 2000). Less expensive materials such as locally-sourced
waste fines from mines or volcanic ash (Longman et al., 2020) should be considered if their heavy metal content is low, but
the choice of treatment material should be considered together with the vegetation and the native minerals. Application of
magnesium-rich materials (e.g. olivine), for example, may help reverse Mg deficiency in *Pinus sylvatica* and *Picea abies* as
dolomite has done (Huettl and Zoettl, 1993), but some other tree species, such as *Acer saccharum*, have a higher demand for
calcium than for magnesium (Long et al., 2009). The treatment of ecologically sensitive catchments always requires caution
as some species, such as *Sphagnum* mosses and lichens, may respond poorly to treatment (Traaen et al., 1997).
Finally, we consider integration of ERW treatments with forest management practices. Dominant CDR pathways
depend on biogeochemical cycling which in turn depends on the life cycle of the dominant trees. For example, base cation
export and therefore $CO_2$ consumption temporarily increases following clear-felling, then decreases while trees are young and
growing due to base cation uptake, and remains low after trees mature due to nutrient recycling (Balogh-Brunstad et al., 2008).
These dynamics may be less obvious in forests which are not clear-felled; *Acer saccharum* forests are often thinned and retain
a canopy as the seedlings are adapted to shade. Individual *Acer saccharum* trees can live for over 300 years, growing relatively
slowly for the first 40 years and attaining maximum height during the first 150 years (Godman et al., 1990). One may expect
to maximise wood production of growing trees with ERW treatments meeting or exceeding the forest demand for previously
limited nutrients such as calcium, which would also minimise soil respiration if the trees allocate less carbon to roots.
Treatments could be repeated as necessary to meet the nutritional needs of sensitive trees or to maintain high $CO_2$ consumption.
Conversely, rising soil pH may not suit some species. For example, *Acer saccharum* normally grows in organic-rich soils with
pH under 7.3 (Godman et al., 1990) and its growth may be hindered at higher pH following large treatments. Outside the main
tree growth phase, and in forests without responsive tree species, $CO_2$ consumption could become the dominant GHG response
to ERW treatments depending on the extent to which it is counteracted by DOC export as soil pH rises (Johnson et al., 2014)
and decomposition rates and fluxes rise (Lovett et al., 2016). Site-specific research is required to determine the optimum
dosage, timing, efficacy and suitability of ERW treatments on acid-impacted forests.

**Appendix A  Contributions of rain/snow precipitation to streamwater chemistry**
We estimated the contribution of rain/snow (Likens, 2016b, a) relative to all other sources using a previously published mixing
model (Négrel et al., 1993). We assume all $Cl^-$ in the water is from rain/snow, noting however that this common treatment of
Cl as an unreactive tracer is not always justified (Lovett et al., 2005). We calculate the contribution of precipitation to the
streamwater ($\alpha_{rain}$) using Na and Cl, which are less affected by nutrient cycling and adsorption than other major ions (Négrel
et al., 1993):
$$\alpha_{rain,Na}(t) = \frac{\left[\frac{Cl}{Na}\right](stream,t)}{\left[\frac{Cl}{Na}\right](rain,t)} ,$$  (A1)

To account for attenuation of the rain/snow precipitation leaching through the soil, Cl/Na and $HCO_3$/Na at any given time ($t$)
are means from the previous three months. We estimate the contribution of rain/snow to other ions such as $HCO_3^-$ in the
streamwater as follows:
$$\alpha_{rain,HCO3}(t) = \alpha_{rain,Na}(t) \times \frac{\left[\frac{HCO3}{Na}\right](stream,t)}{\left[\frac{HCO3}{Na}\right](rain,t)} ,$$  (A2)

## Appendix B  Fraction of calcium derived from wollastonite

We applied an existing two-component mixing model (Peters et al., 2004):

$$\mathbf{X}_{Ca}(t) = \left[ \frac{\left( \left(\frac{^{87}Sr}{^{86}Sr}\right)_{post} - \left(\frac{^{87}Sr}{^{86}Sr}\right)_{pre} \right)\left(\frac{Sr}{Ca}\right)_{pre}}{\left( \left(\frac{^{87}Sr}{^{86}Sr}\right)_{post} - \left(\frac{^{87}Sr}{^{86}Sr}\right)_{pre} \right)\left(\frac{Sr}{Ca}\right)_{pre} + \left( \left(\frac{^{87}Sr}{^{86}Sr}\right)_{Wo} - \left(\frac{^{87}Sr}{^{86}Sr}\right)_{pre} \right)\left(\frac{Sr}{Ca}\right)_{Wo}} \right], \tag{B1}$$

where pre-app and post-app refer to pre-application and post-application streamwater concentrations and Wo refers to wollastonite. The Sr data (Blum, 2019) have been extended through 2015 (**Fig. S1a**). See Supplementary Information for further discussion of the use of strontium and its isotopes as tracers of $Ca^{2+}$ provenance.

## Code availability

The aqueous geochemistry software PHREEQC software, along with documentation, is freely available from the USGS website (https://www.usgs.gov/software/phreeqc-version-3). MATLAB® may be purchased from the MathWorks website (https://uk.mathworks.com/products/matlab.html). Our MATLAB code and scripts used for this project are provided in a supplementary .zip file, without guarantees that these will run with MATLAB versions other than R2016a or on non-Linux operating systems.

## Data availability

Our data are available from the Long Term Ecological Research (LTER) Network Data Portal.  This public repository can be accessed via the Hubbard Brook Ecosystem Study website: https://hubbardbrook.org/d/hubbard-brook-data-catalog
See Supplement for a full list of filenames, package IDs, DOIs and access dates.

## Author contributions

All authors contributed to project conceptualization and interpretation of model results. L.L.T. undertook model simulations and data analysis.  L.L.T. and D.J.B. drafted the manuscript with edits and revisions from all authors. C.T.D. designed the wollastonite watershed study, provided data and observations for model simulations.  J.D.B. provided strontium isotope datasets. P.M.G. provided soil respiration, nitrous oxide and methane flux data.

**Competing interests**
The authors declare that they have no conflict of interest.
**Disclaimer**
**Acknowledgements**
L.L.T. and D.J.B. gratefully acknowledge funding from the Leverhulme Trust through a Leverhulme Research Centre Award
(RC-2015-029). This manuscript is a contribution of the Hubbard Brook Ecosystem Study. Hubbard Brook is part of the Long-
Term Ecological Research (LTER) network, which is supported by the National Science Foundation (DEB-1633026). L.L.T.
thanks Ruth Yanai for a helpful discussion about vegetation, Fred Worrall for advice on flow adjustment and flux calculation,
Peter Wade for advice on the initial PHREEQC setup and Andrew Beckerman and Evan DeLucia for constructive criticism
and advice on statistical modelling. We are grateful to Gregory Lawrence for information about applying lime treatments to
the Appalachian Trail corridor, to Lisa Martel for providing the locations of the trace gas sampling sites and to Habibollah
Fahkraei for creating the watershed map with weir and trace gas sampling locations in Fig. 1. We are grateful for comments
from W. Brian Whalley and editor Tyler Cyronak, and for reviews from Morgan Jones and an anonymous referee which led
to major improvements in this manuscript.

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
