# Peer review of "Increased carbon capture by a silicate-treated forested watershed affected by acid deposition"

_Biogeosciences, 2020_

## Referee Comment (RC1) · Anonymous Referee #1 · 21 Aug 2020

Summary: Taylor et al. present results from an enhanced rock weathering (ERW) field experiment in Hubbard Brook Experimental Forest in the northeastern United States. The authors show observational and modeling evidence in support of sustained carbon dioxide removal for 15 years following the application of silicate minerals to the the experimental plots in 1999. Overall, I find the observational and technical pieces of this manuscript to be very strong. I also found this manuscript difficult to read. I believe the authors could improve the readability, and likely the impact, of the manuscript by revising the structure and flow of the manuscript. Currently, there are an extensive number of equations, missing topic sentences, and redundant sections. These all need to be edited to improve the manuscript. I have tried to highlight some examples below.

General comments:

[Figure]

Please streamline Secs. 2.2.3-2.2.5. After reading them many times, it is still very confusing which equations were used in the modeling, and which are there merely for background context.

Please use math fonts to better differentiate between text and equations. It is very hard to follow the train of logic in the manuscript, which employs 16 equations, without appropriate fonts.

This manuscript would benefit from a table listing all model variables presented in the manuscript with descriptions and units. It is too difficult to keep track of all variables, especially without the use of Math font currently. Also, all model variables need to be used consistently throughout the manuscript. For instance, "X" is used in Eq. 9 and "X_Ca" is used in Eq. 10.

Please embed figures and tables in the appropriate positions in the manuscript, not at the end. This greatly facilitates comprehension of the non-typeset document by reviewers.

Please make arrangements to make the Matlab scripts publicly available, via Github, as a series of supplemental files to the manuscript, or through some other appropriate means. Doing so improves the reproducibility of the science, and allows others to access them without needing to make a "request" (as indicated in the manuscript).

This manuscript is missing a study site figure (probably as Fig. 1) that orients readers to the HBEF and the study and control watersheds.

Please do not reference equations that have yet to be presented in the manuscript (e.g. Eq. 13, L128).

Specific comments:

There are two Sec 2.1.1: Site description and Treatment description. Please correct.

Sec 2.1.1 (Site description): Watershed W1 is never introduced. It needs to be introduced here prior to mention of its flow rates (L74)

I find the transition between Secs. 2.2 and 2.3 to be difficult to follow. Sec. 2.2 presents the modeling approach and the first sentence in Sec 2.3 begins talking about wood production. Please provide some introductory material in Sec. 2.3 prior to discussing the details of the GHG calculations

Inline calculations (e.g. L195) are very difficult to follow and hinder comprehension. Please consider alternative ways to deliver this information to readers.

Secs. 3.2 and 3.3 are essentially sensitivity analyses of the model to different assumptions or scenarios. As such, I think Figs. 2 and 3 could be placed in the SI in order to keep the main figures focused solely on the observational results of the ERW experiment (or model results of the observations)

Sec. 3.5 is identical to Sec. 3.3 (unless I am missing something). Please remove.

Sec 5: This section does not add any new information to the manuscript, especially since key findings were reviewed in Sec. 4. Please remove.

Figs. 1 and 2 are too small to be easily readable. Please enlarge.

Figs. 1-3: The dashed lines representing the treatment should be identified in the figure captions.

Fig. 4: Is time-integrated $CO_2$ flux shown on the y-axis (as implied by L 324)? If so, please correct the y-axis label accordingly.

Table 2/Fig. 5: I find the terminology and axis references incredibly confusing. Please use alternate language that more clearly indicates whether the total greenhouse gas budget has increased or decreased.

Fig. 5: missing panel captions (e.g., "a)", "b)", etc.). Also the caption is excessively long and needs to be shortened.

Fig. 5: I could not find a reference to this figure in the text. Please add a reference. Figure axis text needs to be enlarged across all figures.

[Figure]

---

## Referee Comment (RC2) · Morgan Jones (Referee) · 23 Sep 2020

Overall, I thought this paper was excellent. The manuscript is polished, thorough, and well structured in way that presents a data-heavy study in a concise manner. Testing ways to remove carbon dioxide from the atmosphere is critical for mitigating the response to anthropogenic climate change, which makes this paper of particular significance. I only have one query which I would like some comment on in the discussion, with a couple of minor comments. I recommend that the paper is accepted after these are addressed.

Discussion: The long term efficacy of carbon capture and storage, both in geological and modern examples, seems to hinge on whether organic (via biomass) or inorganic

(via carbonate) carbon are the dominant sinks for increases in atmospheric CO2. Given that this study concludes that uptake into biomass is an important factor for carbon storage in the catchments, what does this mean for the ability of experiments such as this to function on longer time scales (i.e. >100 years). Is this a one-off procedure that can be implemented on a catchment, or can it be repeated with a minimum repose time? Will the draw down via organic and inorganic pathways change with repeated treatments perhaps? I know this is going to be speculative, but I think it would be beneficial for the authors to share their thoughts on how this may be able to be integrated into long term catchment management strategies.

Minor comments:

Line 30: 71 degrees west, rather than -71 degrees east

Line 120: Repetition of "Mohseni and Stefan"

Line 127: What does "mm/time" mean?

Line 192: Repetition of "Battles et al."

Line 350: Replace "3.4 4" with "3.44"

Line 418: A possibility for a low cost alternative to wollastonite could be volcanic ash (see e.g. Longman et al., 2020; https://doi.org/10.1016/j.ancene.2020.100264), particularly in catchments with volcanic deposits nearby.

---

## Author Comment (AC1) · 5 Oct 2020

Reviewer 1: Summary: Taylor et al. present results from an enhanced rock weathering (ERW) field experiment in Hubbard Brook Experimental Forest in the northeastern United States. The authors show observational and modeling evidence in support of sustained carbon dioxide removal for 15 years following the application of silicate minerals to the the experimental plots in 1999. Overall, I find the observational and technical pieces of this manuscript to be very strong. I also found this manuscript difficult to read. I believe the authors could improve the readability, and likely the impact, of the manuscript by revising the structure and flow of the manuscript. Currently, there are an extensive number of equations, missing topic sentences, and redundant sections. These all need to be edited to improve the manuscript. I have tried to highlight some

examples below.

Response: We agree with the reviewer's comments and agree that the paper will be improved if they are attended to. We are particularly grateful that the reviewer noticed that the text of Sec. 3.3 had been copied into Sec. 3.5, which should contain text related to Figure 5 as we state below.

Reviewer 1: General comments: Please streamline Secs. 2.2.3-2.2.5. After reading them many times, it is still very confusing which equations were used in the modeling, and which are there merely for background context.

Response: We will revisit this text and try to make it more clear.

Reviewer 1: Please use math fonts to better differentiate between text and equations. It is very hard to follow the train of logic in the manuscript, which employs 16 equations, without appropriate fonts.

Response: According to the Biogeosciences guidelines, mathematical variables (other than chemical species) should be displayed in italics and we had not done this. It is not clear which math fonts (if any) are preferred by Biogeosciences, but in any case we are happy to comply.

Reviewer 1: This manuscript would benefit from a table listing all model variables presented in the manuscript with descriptions and units. It is too difficult to keep track of all variables, especially without the use of Math font currently. Also, all model variables need to be used consistently throughout the manuscript. For instance, "X" is used in Eq. 9 and "X_Ca" is used in Eq. 10.

Response: We agree that a table listing the variables in Equations 1 through 14 would be useful. The variables in Equations 15 and 16 are already in an existing table but were not in italics as they should have been given journal guidelines.

Reviewer 1: Please embed figures and tables in the appropriate positions in the manuscript, not at the end. This greatly facilitates comprehension of the non-typeset

document by reviewers.

Response: We are happy to embed figures within the text if this is still allowed at this stage.

Reviewer 1: Please make arrangements to make the Matlab scripts publicly available, via Github, as a series of supplemental files to the manuscript, or through some other appropriate means. Doing so improves the reproducibility of the science, and allows others to access them without needing to make a "request" (as indicated in the manuscript).

Response: We will make the MATLAB scripts available such that users will not need to contact us. These scripts were not originally intended for public dissemination; they include some existing scripts from earlier work requiring (for example) that extra header lines are added to the publicly available data files. This is less than ideal for anyone wishing to reproduce or extend our analyses, but we will provide a README file explaining these things along with the altered datafiles and the code.

Reviewer 1: This manuscript is missing a study site figure (probably as Fig. 1) that orients readers to the HBEF and the study and control watersheds.

Response: We are happy to add a new Figure 1 showing the study site.

Reviewer 1: Please do not reference equations that have yet to be presented in the manuscript (e.g. Eq. 13, L128).

Response: Agreed. Reviewer 1: Specific comments:

Reviewer 1: There are two Sec 2.1.1: Site description and Treatment description. Please correct.

Response: Yes, the second one should be Sec. 2.1.2 Treatment description

Reviewer 1: Sec 2.1.1 (Site description): Watershed W1 is never introduced. It needs to be introC2 BGD Interactive comment Printer-friendly version Discussion paper duced here prior to mention of its flow rates (L74)

Response: Watershed 1 was introduced after watershed 6. We have reversed this so watershed 1 is now introduced first.

Reviewer 1: I find the transition between Secs. 2.2 and 2.3 to be difficult to follow. Sec. 2.2 presents the modeling approach and the first sentence in Sec 2.3 begins talking about wood production. Please provide some introductory material in Sec. 2.3 prior to discussing the details of the GHG calculations

Response: The material currently in Sec. 2.3.4 (Greenhouse gas budget for a treatment) should probably be moved to the beginning of Sec. 2.3. With slight alterations it should actually provide the introductory material the reviewer is asking for.

Reviewer 1: Inline calculations (e.g. L195) are very difficult to follow and hinder comprehension. Please consider alternative ways to deliver this information to readers.

Response: The inline calculation can be moved.

Reviewer 1: Secs. 3.2 and 3.3 are essentially sensitivity analyses of the model to different assumptions or scenarios. As such, I think Figs. 2 and 3 could be placed in the SI in order to keep the main figures focused solely on the observational results of the ERW experiment (or model results of the observations)

Response: We are happy to move Figs 2 and 3 to the supplementary information.

Reviewer 1: Sec. 3.5 is identical to Sec. 3.3 (unless I am missing something). Please remove.

Response: We are grateful to the reviewer for catching this mistake. Sec. 3.5 is where the logistical penalties and greenhouse gas balance are discussed but the text seems to have been obliterated during the final stages of editing.

Reviewer 1: Sec 5: This section does not add any new information to the manuscript, especially since key findings were reviewed in Sec. 4. Please remove.

Response: Agreed.

Reviewer 1: Figs. 1 and 2 are too small to be easily readable. Please enlarge.

Response: Agreed.

Reviewer 1: Figs. 1-3: The dashed lines representing the treatment should be identified in the figure captions.

Response: Agreed.

Reviewer 1: Fig. 4: Is time-integrated CO2 flux shown on the y-axis (as implied by L 324)? If so, please correct the y-axis label accordingly.

Response: Agreed, the word "flux" should be in the Y axis label.

Reviewer 1: Table 2/Fig. 5: I find the terminology and axis references incredibly confusing. Please use alternate language that more clearly indicates whether the total greenhouse gas budget has increased or decreased.

Response: This table and figure may have been substantially less confusing if the proper text of Sec. 3.5 had been included in the manuscript. Also, re-examination of Fig. 5b suggests it ought to be overhauled to make clear the linkage with Table 2 and also whether the components and total budget are carbon sources or sinks.

Reviewer 1: Fig. 5: missing panel captions (e.g., "a)", "b)", etc.). Also the caption is excessively long and needs to be shortened.

Response: There are several unnecessary sentences in the caption, but on reconsideration we also suggest that Fig 5a and Fig 5b form separate figures (with Fig 5b overhauled as noted above). Panels c and d add little to the story and we think they should be jettisoned. If these things are done then the caption(s) would be considerably shortened and in our opinion less confusing for readers.

Reviewer 1: Fig. 5: I could not find a reference to this figure in the text. Please add a

reference.

Response: The references to this figure were in the missing text of Sec. 3.5.

Reviewer 1: Figure axis text needs to be enlarged across all figures.

Response: We have appended new versions of all figures in the main manuscript, but are happy to make further changes as required. In particular, we now have a site map as Figure 1, and Figure 5 now has only one panel which is entirely overhauled. It shows the variable names appearing in the old Table 2, Equations 15 and 16, and in the revised text, as well as the meanings of those variables so that the figure is comprehensible without reading the text. We think it is now obvious that the treated watershed is a net greenhouse gas sink. We will nevertheless be happy to revise them again as required.

Please also note the supplement to this comment:
https://bg.copernicus.org/preprints/bg-2020-288/bg-2020-288-AC1-supplement.pdf
* * *
![Map showing the location of sampling sites and the Hubbard Brook Experimental Forest. Watersheds labeled W1–W6, with red dots marking microbial and trace gas plots, green triangles marking weirs, and blue lines marking streams. Inset shows New Hampshire State location in the USA.]

**Hubbard Brook Experimental Forest**

New Hampshire State

W3

W2

W1

W6

W5

W4

N

Meters
0   125   250      500

● Microbial and trace gas plots
▼ Weirs
— Streams

**Fig. 1.** Location of the sampling sites and the Hubbard Brook Experimental Forest.

[Figure]

**Fig. 2.** Inorganic CO2 capture at the Hubbard Brook Experimental Forest.

[Figure]

**Fig. 3.** Long-term soil respiration responses to wollastonite treatment at Hubbard Brook Experimental Forest.

[Figure]

**Fig. 4.** Carbon penalties for the wollastonite treatment.

**GHG Source  GHG Sink**

| | |
|---|---|
| **LOGPEN** | Logistical penalty given actual transport |
| **LOGPEN** | Logistical penalty with shorter transport |
| **△DOC** | $CO_2$ emissions from exported DOC  (range: –0.203 to –0.000) |
| **△NO3NO2** | $N_2O$ emissions from exported $NO_3$  (range: –0.071 to –0.016) |
| **△CONS** |  $CO_2$ consumption  (range: 0.025 to 0.129) |
| **△CH4** | Soil $CH_4$ sink  (range: 0.015 to 0.029) |
| **△SRESP** | Soil respiration reduction |
| **△wood** | Wood production (10 years, Battles et al 2014) |
| **△GHG** | Actual net GHG sink  **SINK** |
| **△GHG** | Net GHG sink with shorter transport  **SINK** |

Cumulative GHG sink end 2014 (t $CO_{2e}$ ha$^{-1}$)

**Fig. 5.** Carbon responses for the wollastonite treatment.

[Figure]

Fig. 6. Projected CO2 consumption following higher-dosage treatments

**Supplement:**

**Supplementary Information**

**Increased carbon capture by a silicate-treated forested watershed affected by acid deposition**

Lyla L. Taylor[1*], Charles T. Driscoll[2], Peter M. Groffman[3], Greg H. Rau[4], Joel D. Blum[5] and David J. Beerling[1]

[1]Leverhulme Centre for Climate Change Mitigation, Department of Animal and Plant Sciences, University of Sheffield, Sheffield S10 2TN, UK

[2]Department of Civil and Environmental Engineering, 151 Link Hall, Syracuse University, Syracuse, NY 13244, USA

15   [3]City University of New York, Advanced Science Research Center at the Graduate Center, New York, NY 10031 and Cary Institute of Ecosystem Studies, Millbook, NY 12545 USA

[4]Institute of Marine Sciences, University of California, Santa Cruz, CA 95064 USA

[5]Department of Earth and Environmental Sciences, University of Michigan, Ann Arbor, MI 48109, USA

*Correspondence to*: Lyla L. Taylor (L.L.Taylor@sheffield.ac.uk)

**1 Geochemistry**

**1.1 Carbonic acid equilibria**

In an open system where gaseous $CO_2$ is not limited, such as water in contact with the air, Henry's Law means that carbonic acid is linearly proportional to the partial pressure of $CO_2$ gas ($pCO_2(g)$):

$$[H_2CO_3^*] = K_H \times pCO_2(g) \tag{S1}$$

where hydrated $CO_2$ ($CO_2(aq)$) and carbonic acid ($H_2CO_3$) are combined as $H_2CO_3^*$. Speciation of carbonate and bicarbonate ions is given by equilibria depending on $[H_2CO_3^*]$ and on $[H^+]$, where $pH = -\log_{10}[H^+]$:

$$[HCO_3^-] = [H_2CO_3^*] \times [H^+] \times K_1 \tag{S2}$$

$$[CO_3^{2-}] = [HCO_3^-] \times [H^+] \times K_2 = [H_2CO_3^*] \times [H^+]^2 \times K_1 \, K_2 \tag{S3}$$

Similar equilibrium expressions apply to organic acids.

**1.2 Alkalinity, major ions and acids**

Relationships between $SO_4^{2-}$, $NO_3^-$ and organic and carbonic acids can be illustrated via expressions for alkalinity. Following Stumm and Morgan (Stumm and Morgan 1996), a simple expression for freshwater alkalinity is given by:

$$Alk = [HCO_3^-] + 2[CO_3^{2-}] + [OH^-] - [H^+] \tag{S4a}$$

Equation S4a excludes non-carbonate alkalinity found in New Hampshire rivers (Hunt, Salisbury, and Vandemark 2011), but can be modified to include species related to the organic acid $H_3A$ from Equation 1d:

$$Alk = [HCO_3^-] + 2[CO_3^{2-}] + [OH^-] - [H^+] + [H_2A^-] + 2[HA^{2-}] + 3[A^{3-}] \tag{S4b}$$

A general alkalinity expression including non-carbonate alkalinity (Stumm and Morgan 1996) is:

$$Alk = [HCO_3^-] + 2[CO_3^{2-}] + [OH^-] - [H^+] + [NH_3] + [HS^-] + 2[S^{2-}] + [H_3SiO_4^-] + 2[H_2SiO_4^{2-}] + [B(OH)_4^-]$$

$$+ [A^-] + [HPO_4^{2-}] + 2[PO_4^{3-}] - 2[H_2PO_4^{2-}] \tag{S4c}$$

where A is a collective term for organic acids . Boric acid species $[H_3BO_3]$ and $[B(OH)_4^-]$ are included in seawater.

In practice, $CO_3^{2-}$ is usually neglected in $CO_2$ consumption calculations (Raymond 2017) because it is several orders of magnitude smaller than $HCO_3^-$ within the typical pH range of most natural waters [4.5-9]. In this pH range, an alternative definition of alkalinity, based on major ions (Stumm and Morgan 1996), arises from the requirement for electrical neutrality of the solution:

$$Alk = 2([Ca^{2+}] + [Mg^{2+}] - [SO_4^{2-}]) + [K^+] + [Na^+] + [NH_4^+] - [NO_3^-] - [Cl^-] - [F^-] \tag{S5}$$

The anions and cations in eqn. S5 are associated with fully dissociated acids and bases in the pH range of natural waters, as described above. Consequently, they strongly control streamwater chemistry and pH ($-\log_{10}[H^+]$) because S4a or S4b must equal S5. Speciation of organic and carbonic acids depends on pH, so these weak acids react to conditions created by the major ions. Eqns. 4 and 5 suggest that changes to any major ion in Eqn. S5 will result in concomitant adjustments in the ions

50  of 4. In turn, Eqns 4a and 4b imply that, given the major ions, ignoring the organic acid could increase the calculated bicarbonate concentrations.

**1.2 Carbonate mineral weathering**

It has been common to treat forests with carbonate minerals such as $CaCO_3$ (calcite or aragonite) or $CaMg(CO_3)_2$ (dolomite) to mitigate the effects of anthropogenic acidification. Here we present $CaCO_3$ weathering equations analogous to our
55  wollastonite equations (1–4, main manuscript):

$$CaCO_3 + H_2O + CO_2 \leftrightarrows Ca^{2+} + 2HCO_3^- \tag{S6}$$

$CaCO_3$ weathering by carbonic acid results in two bicarbonate ions, one from the $CaCO_3$ and one from the atmosphere. It
60  will effectively raise the pH of the solution, but compared to wollastonite weathering (1, main manuscript), it draws down half the $CO_2$ from the atmosphere. When a new $CaCO_3$ molecule is precipitated by an organism or by evaporation, the second bicarbonate ion will be released as $CO_2$ to maintain charge balance of the solution.

$$CaCO_3 + H_2SO_4 \rightarrow Ca^{2+} + H_2O + CO_2 + SO_4^{2-} \tag{S7}$$
65  $$CaCO_3 + 2HNO_3 \rightarrow Ca^{2+} + H_2O + CO_2 + 2NO_3^- \tag{S8}$$
$$CaCO_3 + 2H_3A \rightarrow Ca^{2+} + H_2O + CO_2 + 2H_2A^- \tag{S9}$$

Following $CaCO_3$ weathering by other acids, the resulting acid-derived anions will prevent the formation of bicarbonate and result in release of $CO_2$ derived entirely from the weathered $CaCO_3$. Bicarbonate formation requires neutralization of these
70  acid-derived anions.

Dolomite weathering obeys a similar set of equations. Ion production is effectively doubled compared to $CaCO_3$ weathering, as each dolomite molecule is equivalent to one $CaCO_3$ plus one $MgCO_3$ molecule:

$$CaMg(CO_3)_2 + 2H_2O + 2CO_2 \leftrightarrows Ca^{2+} + Mg^{2+} + 4HCO_3^- \tag{S10}$$
75  $$CaMg(CO_3)_2 + 2H_2SO_4 \rightarrow Ca^{2+} + Mg^{2+} + 2H_2O + 2CO_2 + 2SO_4^{2-} \tag{S11}$$
$$CaMg(CO_3)_2 + 4HNO_3 \rightarrow Ca^{2+} + Mg^{2+} + 2H_2O + 2CO_2 + 4NO_3^- \tag{S12}$$
$$CaMg(CO_3)_2 + 4H_3A \rightarrow Ca^{2+} + Mg^{2+} + 2H_2O + 2CO_2 + 4H_2A^- \tag{S13}$$

Equations **S7** and **S10** show that carbonate mineral weathering by carbonic acid (eqns **S6** and **S10**) draws down half the
80  atmospheric $CO_2$ of silicate weathering per cations released. Because silicate minerals do not contain carbon, they cannot release $CO_2$ when weathered. Silicate minerals are therefore of greater interest than carbonate minerals in the context of CDR.

**1.3 Use of strontium isotopes as calcium tracers**

Our use of Eqn. 10 (Methods) to determine the fraction $X_{Ca}$ originating from wollastonite (**Fig. S1a**) follows the widely-held assumption that strontium (Sr) is an isotopic tracer of calcium provenance (Nezat, Blum, and Driscoll 2010; Peters et al. 2004). Being divalent cations in the same group in the periodic table, Sr and Ca have similar chemical behaviour and only slightly differing atomic radii. Strontium can therefore replace calcium in a variety of minerals, in particular plagioclase and apatite. Incorporation of Sr within mineral lattices in place of Ca is favoured by eight-fold coordination with oxygen (Capo, Stewart, and Chadwick 1998).

Rubidium (Rb) substitutes for K in minerals, but because $^{87}Rb$ decays to $^{87}Sr$, and because $^{86}Sr$ and $^{87}Sr$ are both stable isotopes, the ratio $^{87}Sr/^{86}Sr$ is distinctive for rocks of similar age and Rb/Sr ratio. The $^{87}Sr/^{86}Sr$ ratio can also trace the provenance of Sr in soil, pore waters, pedogenic carbonates and other secondary minerals because these pools have the same isotopic composition as the source material(Capo, Stewart, and Chadwick 1998). Isotopic fractionation is insignificant during chemical weathering, physical erosion and atmospheric transport (Bain and Bacon 1994) as well as biological processes (Capo, Stewart, and Chadwick 1998) including nutrient uptake.

Changes in Ca/Sr ratios due to any of the above processes are minor (Blum et al. 2002), but some processes do lead to partitioning between Ca and Sr which may need to be accounted for (Pett-Ridge, Derry, and Barrows 2009). For example, Sr is more strongly retained in soils than Ca (Capo, Stewart, and Chadwick 1998), and partitioning also occurs during biogeochemical cycling processes such as nutrient uptake and within-plant transport (Pett-Ridge, Derry, and Barrows 2009). In a study of apatite weathering at Hubbard Brook, the two molar ratios $^{87}Sr/^{86}Sr$ and Ca/Sr were found to be distinctive for atmospheric inputs, bedrock silicates and apatite (**Fig. S1b**).

Comparison with these three end-members suggested that streamwaters, pore waters and particularly the soil exchange pool held little apatite-Ca (Blum et al. 2002). $^{87}Sr/^{86}Sr$ was found to be similar in the leaves of different species of trees, ferns, soil exchangers, soil and stream waters, and soil apatite, while Ca/Sr was distinctive (Blum et al. 2002). Together, these two ratios suggested that apatite was an important source of calcium for the dominant forest trees other than Ca-sensitive *Acer saccharum* in the treated watershed prior to the wollastonite treatment (Blum et al. 2002).

The wollastonite used in our experiment was formed by contact metamorphism between silica and limestone (Virta 2000). Its $^{87}Sr/^{86}Sr$ ratio is therefore typical of limestone (Capo, Stewart, and Chadwick 1998) and consequently easily distinguished from other Ca sources (Blum et al. 2002) at the HBEF. Like the apatite found in these soils (Blum et al. 2002), it has a high Ca/Sr ratio, whereas pre-treatment streamwater Ca/Sr was considerably lower (Peters et al. 2004). Peters et al (Peters et al. 2004) showed that the applied wollastonite was also isotopically distinct from pretreatment streamwater and that streamwater ratios during the first year after treatment could be explained by mixing between these two end-members (Fig. S4b). Ca/Sr during the infiltration period 3+ years post-treatment appears to show a slight tendency towards apatite (**Fig. S1b,c**) but this is expected as ~35% of streamwater Ca was derived from apatite prior to treatment (Blum et al. 2002).

**(a)**

[Figure]

**(b)**                                                                                  **(c)**

[Figure]

125

**Figure S1. Strontium as a tracer.** (a) Contribution of wollastonite Ca to streamwater $Ca^{2+}$ in the treated watershed, using the mixing model
of Peters et al (Peters et al. 2004) and showing the conceptual model of Nezat et al (Nezat, Blum, and Driscoll 2010). This mixing model
assumes that mixing is linear between two endmembers. According to the conceptual model, wollastonite-derived Ca did not appear in
streamwater until three years had elapsed. Subsequent work has supported this conceptual model (Shao et al. 2016). (b) Ca/Sr and Sr isotopic

130     ratios change following treatment, suggesting a change in provenance toward wollastonite (Peters et al. 2004) rather than toward other
calcium sources (Blum et al. 2002). (b) Blum et al (Blum et al. 2002) found that ~35% of the Ca in pretreatment streamwater originated
from apatite weathering; this signal is still apparent during the infiltration period (>3 years post-treatment).

**2 CO₂ consumption at Hubbard Brook**

135    When calculated using bicarbonate and carbonate ions, $CO_2$ consumption ($CO_{2,HCO3}$) depends strongly on the pH of the streamwater, which was similar for our two watersheds prior to treatment (**Fig. S2a**). After treatment, pH rose sharply in the treated watershed but soon approached values only slightly higher than the reference watershed.

        $CO_2$ consumption is directly proportional to flow rates, which are slightly lower in the  treated watershed (**Fig. S2b**),
140    particularly in late summer (**Fig. S2c**). Consideration of the total rolling annual streamflow, where each point is the sum of the flow over the preceding 365 days, shows that flow was higher in the reference watershed prior to treatment and for several years following treatment (**Fig. S2d**).

[Figure]

145    **Figure S2.  Comparison of the treated and reference watersheds since 1991.** (a) Streamwater pH, (b) total daily flow, (c) ratio of treated/reference daily flow on a monthly basis displayed as boxplots (center line median, box limits upper and lower quartiles, whisker length 1.5×interquartile range, including all outliers (•) and (d) rolling annual streamflow, where each point is the sum of the preceding 365 days.  Note that the reference watershed dataset (Driscoll 2016b) extends into the 1980s whereas the treated watershed dataset (Driscoll 2016a) does not.

150

**2.1 Contribution of rainwater to CO₂ consumption fluxes**

Calculation of rainwater contributions is described in the Methods. **Fig. S3** shows the rainwater contribution to bicarbonate, while **Fig. S4** shows rainwater contributions to $Na^+$, $Ca^{2+}$, $SO_4^{2-}$, and $NO_3^-$. Note that the nitrate spikes discussed in the main text (2012–2015) are not contributed by rainwater.

155

Many authors correct cation-based CO₂ consumption for $SO_4^{2-}$ originating from weathering (CO$_{2,ions}$, see Methods in main text), but at HBEF $SO_4^{2-}$ is largely derived from rainwater. In principle, all the $SO_4^{2-}$ at the HBEF is from acid deposition via rainwater, but some had been sorbed to soil oxides and was desorbed following treatment. The simple rainwater correction shown in **Fig. S4** may not account for $SO_4^{2-}$ desorbed from soil oxides (**Fig. S4**), but this legacy $SO_4^{2-}$ still originates from
160 atmospheric deposition. Because total $SO_4^{2-}$ will reduce CO₂ consumption (CO$_{2,HCO3}$ or CO$_{2,ions}$) along with marine alkalinity and therefore storage of bicarbonate in the water column, we do not apply any rainwater correction to $SO_4^{2-}$ when calculating CO$_{2,ions}$.

[Figure]

165

**Figure S3. Contribution of rainwater in treated (top) and reference (bottom) watersheds.** In each panel, the bottom of the envelope corresponds to the higher atmospheric CO₂ simulations. A three-point smoothing has been applied to these curves.

[Figure]

170

**Figure S4. Fractional contribution of rainwater to Na⁺, Ca²⁺, SO₄²⁻, and NO₃⁻** for our baseline case in the treated watershed. The bottom and top axes for each panel represent 0 (no rainwater contribution) and 1 (all from rainwater) respectively. The time of treatment is indicated with a vertical dotted line.

**2.2 Comparison with literature**

175 Our annual $CO_2$ consumption estimates are within the ranges of most previous estimates for Hubbard Brook watersheds (**Table S1**). We estimate that 2% of the applied Wo-Ca was exported during the first six years following treatment, and 4% was exported by the end of year 11. These are within the range of prior published studies (Shao et al. 2016; Cho et al. 2012). Our mean treatment-related $CO_2$ consumption rates for the period 2003–2012 were Wo-$CO_{2,HCO3}$=0.004 and Wo-$CO_{2,Ca}$=0.02 mol C $m^{-2}$ $y^{-1}$. The latter corresponds to 43.9 kg Wo-Ca $yr^{-1}$, four-times higher than predicted by Nezat et al (Nezat, Blum, and

180 Driscoll 2010), but similar to results presented by Shao et al (Shao et al. 2016) (49.4 kg Wo-Ca $yr^{-1}$). Our estimated Wo-Ca export rate suggests that increased $CO_2$ consumption following treatment could last for ~140 yrs, i.e., the time taken to deplete the 14 metric tons of calcium applied to the treated watershed.

185 **Table S1**. Estimates of annual $CO_2$ consumption at the treated and reference catchments Hubbard Brook. Where calculated from wollastonite-derived calcium (Wo-Ca) export, perfect efficiency is assumed following 1.

| CO₂ consumption | | | Catchment | Method | Period | Source |
|---|---|---|---|---|---|---|
| t CO₂ ha⁻¹ | g C m⁻² y⁻¹ | mol C m⁻² y⁻¹ | | | | |
| 0.0044 | 0.12 | 0.01 | REF (6) | H⁺ accounting | pretreatment | (Fahey et al. 2005) |
| 0.0009 | 0.024 | 0.002 | REF (6) | Bicarbonate, 5 | pretreatment–2002 | This study |
| 0.0013 | 0.036 | 0.003 | REF (6) | Bicarbonate, 5 | mean, 2003–2014 | This study |
| 0.0004 | 0.012 | 0.001 | Treated (1) | Bicarbonate, 5 | pretreatment | This study |
| | | | | | | |
| 0.0110 | 0.30 | 0.025 | Treated (1) | Wo-Ca export | mean, 1999–2010 | (Schlesinger and Amundson 2018) their SI |
| 0.0048 | 0.133 | 0.011 | Treated (1) | Wo-Ca export | mean, 1999–2010 | (Shao et al. 2016) their Table 1 |
| 0.0048 | 0.132 | 0.011 | Treated (1) | Wo-Ca export | mean 2003–2010 | (Shao et al. 2016) their Table 1 |
| 0.0022 | 0.056 | 0.005 | Treated (1) | Wo-Ca export | mean 2005–2008 | (Nezat, Blum, and Driscoll 2010) |
| 0.0084 | 0.228 | 0.019 | Treated (1) | Wo-Ca, 9 | mean 2003–2014 | This study |
| 0.0018 | 0.048 | 0.004 | Treated (1) | Bicarbonate, 7 | mean 2003–2014 | This study |
| **per unit watershed area** | | | | | | |
| 0.0167 | 0.459 | 0.038 | Treated (1) | Wo-Ca export | year 1 | (Hartmann and Kempe 2008) |
| 0.0260 | 0.703 | 0.059 | Treated (1) | Wo-Ca export | year 1 | (Peters et al. 2004) |
| 0.0216 | 0.586 | 0.049 | Treated (1) | Wo-Ca export | year 1 | (Shao et al. 2016) their Table 1 |
| 0.0229 | 0.625 | 0.052 | Treated (1) | Wo-Ca, 9 | year 1 | This study |
| 0.0048 | 0.132 | 0.011 | Treated (1) | Bicarbonate, 7 | year 1 | This study |
| **per unit streambed area** | | | | | | |
| 1.1003 | 30 | 2.5 | Treated (1) | Wo-Ca export | year 1 | (Hartmann and Kempe 2008) |
| 1.6843 | 45.96 | 3.827 | Treated (1) | Wo-Ca export | year 1 | (Peters et al. 2004) |
| 1.4044 | 38.33 | 3.191 | Treated (1) | Wo-Ca export | year 1 | (Shao et al. 2016) their Table 1 |
| 1.4981 | 40.88 | 3.404 | Treated (1) | Wo-Ca, 9 | year 1 | This study |
| 0.3164 | 8.634 | 0.719 | Treated (1) | Bicarbonate, 7 | year 1 | This study |

**2.3 Sensitivity to other acids**

At Hubbard Brook, $CO_2$ consumption as computed with bicarbonate ($CO_{2,HCO3}$) was reduced due to the presence of environmental acids other than carbonic acid, as discussed in the main text. **Fig. S5** shows the solute concentrations, solute fluxes and $CO_2$ consumption curves for the sulphuric acid (**Fig. S5a,b,c**), nitric acid (**Fig. S5d,e,f**) and organic acid (**Fig. S5g,h,i**) scenarios described in **Section 3.2** of the main text.

195

[Figure]

**Figure S5. Sensitivity of watershed inorganic $CO_2$ capture at Hubbard Brook to environmental change.** Modelled watershed steamwater bicarbonate and corresponding patterns of total $CO_2$ consumption ($CO_{2,HCO3}$ by Eq. 5) and treatment-associated $CO_2$ consumption (Wo-$CO_{2,HCO3}$ by Eq. 7) following (a–c) removal of acid deposition effects ("Low SO4" scenario) or (d–f) removal of transient nitrate spiking ("REF NO3" scenario), and (g–i) sensitivity to the presence or absence of organic acids (OA+ and OA-, respectively). Bicarbonate concentrations (b,e,h) for all scenarios are shown to the same scale. All $CO_2$ consumption curves (c,f,i) are shown to the same scale and were calculated with flow-normalised concentrations and corrected for sparsity of samples (Methods).

200

205

**2.4 Sensitivity to treatment size**

As discussed in the main text **Section 3.3**, **Fig. S6** shows the increases in calcium and bicarbonate concentrations (**Fig. S6a**), rolling annual $CO_2$ consumption (**Fig. S6b**), and cumulative $CO_2$ consumption (**Fig. S6c**) given a ten-fold larger wollastonite treatment.

[Figure]

**Figure S6: Simulated inorganic $CO_2$ capture for a 10-fold higher wollastonite treatment in the Hubbard Brook Experimental Forest.**
(a) Calcium and bicarbonate concentrations, along with observed reference calcium. Calcium is charge-balanced by up to two moles of bicarbonate. (b) $CO_2$ consumption due to this higher treatment (Wo-CO$_{2,Ca}$ and Wo-CO$_{2,HCO3}$) and (c) Cumulative $CO_2$ consumption for both this higher treatment and the actual treatment (Wo-CO$_{2,Ca}$). We have assumed that a 10-fold higher treatment produces 10-fold higher calcium concentrations with no change in sulphate, nitrate or DOC. Simulated HCO$_3^-$ concentration and Wo-CO$_{2,HCO3}$ account for the presence of organic acids (+OA) given observed DOC. All $CO_2$ consumption curves (b,c) were flow-normalised and corrected for sparsity of samples (Methods).

**3 Downstream effects**

**3.1 Degassing in the HBEF streams, rivers and near-shore environments**

If $CO_2$ consumption is determined by bicarbonate ($CO_{2,HCO3}$, 5), it will be subject to substantial temporal and spatial variability due in part to riverine $CO_2$ outgassing linked to geomorphological heterogeneities, seasonality, and changes in connectivity with subsurface water flows (Duvert et al. 2018). Hubbard Brook, for example, is part of the larger Merrimack Watershed that experiences large seasonal $CO_2$ degassing before and after entering the Merrimack estuary (Salisbury et al. 2008).

Progressive degassing begins in the steep headwaters of the HBEF, and continues far downstream of the weirs where our samples were collected. Near the Merrimack river mouth, waters are typically oversaturated with respect to atmospheric $CO_2$ gas in November (~1200 µatm) and undersaturated in July (Salisbury et al. 2008), suggesting seasonal degassing along the length of the river. In the Merrimack estuary, the fresh river water forms a buoyant plume extending into the Gulf of Maine. As a result of wind-driven mixing with the underlying salt water (Chen, MacDonald, and Hetland 2009), the surface water approaches equilibrium with atmospheric $CO_2$ by degassing in November or acting as a $CO_2$ sink in July (Salisbury et al. 2008). Dissolved carbon concentration (DIC) is therefore extremely sensitive to location and timing. In equilibrium with atmospheric $CO_2$, peak DIC and $CO_2$ consumption in the treated watershed decrease by over 70% (not shown).

Degassing in estuaries is inversely related to DIC and buffering capacity (Bauer et al. 2013) and may represent wetland microbial respiration rather than riverine carbon (Cai 2011). For some large, alkaline rivers with high-$pCO_2$ estuaries, degassing represents only a small fraction of the DIC transported to the ocean (Cai 2011). Globally, estuaries export 10% more carbon seaward than to the atmosphere, but continental shelves are undersaturated because transport across the shelves only takes a few months (Bauer et al. 2013). Unfortunately, the complex interplay between aquatic heterotrophs and productivity, and labile dissolved organic carbon (DOC) and nutrient fluxes on carbon cycling in estuaries is not well understood (Bauer et al. 2013; Salisbury et al. 2008).

The amount and duration of DIC storage at sea may depend on poorly understood processes of air-sea gas exchange and remineralisation rates of sinking organic matter (Williamson et al. 2012) as well as regional limitations on nutrient supply to the ocean surface and thus to marine photosynthesis (Williamson et al. 2012; Marinov et al. 2008). ERW with basalt may increase phosporus and iron fluxes to the oceans, but at present there is no protocol for representing the contribution of such nutrient fluxes for carbon accounting.

These processes indicate that streamwater DIC will not reflect carbon sequestration in the ocean water column, and indeed $CO_2$ consumption and longer-term storage are more closely related to streamwater alkalinity (Renforth and Henderson 2017). We have not explicitly addressed degassing in our study, although we have considered effects of weathering by acids other than carbonic acid (Eqns. 2–4 main text), and downstream greenhouse gas emissions associated with nitric and organic acids.

**3.2 Dissolved organic carbon**

The chemistry and isotopic composition of marine DOC indicates it is largely of marine origin (Bianchi 2011), implying net removal of terrestrial DOC before it reaches the high seas. At low latitudes, terrestrial DOC may be oxidised in coastal waters over continental shelves, contributing to $CO_2$ degassing fluxes (Cai 2011). Our pessimistic assumption is that all excess exported DOC (associated with the treatment) is oxidised or respired downstream, leading to $CO_2$ emissions and therefore to a carbon penalty. The relevance of organic acids associated with marine DOC for marine DIC storage is unclear, as is the

interplay between ERW and terrestrial DOC contributions to degassing. However, policy-makers should be aware that DOC-related organic acids can affect total alkalinity measurements in low-pH, organic-rich streamwaters, if carbon-based $CO_2$ consumption ($CO_{2,HCO3}$, 5) is employed for verification.

265

**3.3 Nitrate**

Nitrate export in streamwater is unlikely to affect marine storage because of its low concentrations in seawater far from the coast. At least 80% of inorganic nitrogen is denitrified in coastal waters (Seitzinger and Giblin 1996; Bauer et al. 2013) under low-oxygen conditions caused by eutrophication, but denitrification is associated with greenhouse gas ($N_2O$) emissions

270 (Canfield, Glazer, and Falkowski 2010) which could counteract $CO_2$ sequestration achieved with ERW. This greenhouse gas penalty can be quantified by converting streamwater nitrate to $N_2O$ emissions in rivers and estuaries, and converting $N_2O$ to equivalent carbon dioxide emissions ($CO_{2e}$) as described in Methods. In our study, the absolute magnitude of this penalty (**Fig. 3c** and **Table 2**, main manuscript) exceeded bicarbonate-derived $CO_2$ consumption associated with the wollastonite treatment (Wo-$CO_{2,HCO3}$). In view of these results and the nitrate-treatment interactions observed at Hubbard Brook(Rosi-

275 Marshall et al. 2016), near-shore $N_2O$ emissions could be enhanced by other rock dust treatments on land. Although $CO_{2,HCO3}$ ($CO_2$ consumption calculated with bicarbonate) is sensitive to nitrate, sulphate and organic acids, it cannot fully account for the complex C and N dynamics of nearshore environments.

**4 Forest effects**

**4.1 Soil respiration, other soil greenhouse gas emissions and elevation**

280 Greenhouse gas measurements were made in four vegetation zones which are characterised by different tree species composition and different responses to the wollastonite treatment. We carried out statistical analyses of our measured data (**Fig. S7, S8** and **S9**) as described in the main text. Reduced cumulative soil respiration in the high-elevation hardwood zone in the treated watershed (**Fig. 2**, main text) may be related to lower fine root biomass observed in 2013 compared to 1998 (Fahey et al. 2016) rather than microbial decomposition. Although the treatment resulted in significant increases in hardwood

285 leaf litter decay relative to a nearby control site (Lovett, Arthur, and Crowley 2016) and significant decreases in organic matter and carbon in the Oa horizon relative to pre-treatment (Johnson et al. 2014), it is not clear that these effects were elevation dependent or how the Oa horizon compares with the reference watershed. However, soils were more depleted in Ca at higher elevation than at lower elevations, and the Ca-sensitivity of *Acer saccharum* Marsh. increases with increasing elevation (Minocha et al. 2010).

290 Our monthly measurements showed no significant effects for soil $N_2O$ emissions (**Fig. S8**). Similar to an earlier study comparing the HBEF valley to the highest HBEF elevations (Groffman et al. 2009b), we found that the lower-elevation hardwood soils were stronger sinks for $CH_4$ than higher elevation hardwood soils in the treated watershed (**Fig. S9, S10**). In the reference watershed, however, mid- to high-elevation hardwoods have been smaller sinks than low-elevation hardwoods since 2002 (**Fig. S10**).

295 ### 4.2 Trees as a sink for calcium

We considered the possibility that the extra wood production identified by Battles et al (Battles et al. 2014) could have explained part of the discrepancy in the calcium budget identified by previous authors (Shao et al. 2016). Given ~661 µgCa $g^{-1}$ wood (Arthur, Siccama, and Yanai 1999), this effect amounts to 8.8 mmol Ca $m^{-2}$, well below the missing 1822.5 out of 2570 mmol Wo-Ca $m^{-2}$ applied (Shao et al. 2016). The Wo-Ca budget remains unclosed.

300

[Figure]

**Figure S7. Soil respiration for four vegetation types:** (a) SpruceFir, (b) high-elevation hardwoods, (c) mid-elevation hardwoods, and (d) low-elevation hardwoods. Data are shown for the three chambers per watershed in each forest type at each time. As stated in the main text, the only significant effect was for high-elevation hardwoods. Portions of these data have been published previously (Groffman et al. 2006; Groffman et al. 2009a).

305

[Figure]

310 **Figure S8. Soil N₂O emissions for four vegetation zones:** (a) SpruceFir, (b) high-elevation hardwoods, (c) mid-elevation hardwoods, and (d) low-elevation hardwoods. Data are shown for the three chambers per watershed in each forest type at each time. Values outside the axes limits are: (a) High elevation, August 2003, chamber 3 treated watershed, -1.6502 mmol $N_2O$ m$^{-2}$ month$^{-1}$. (b) SpruceFir, July 2006, chamber 3 reference, 6.0540 mmol $N_2O$ m$^{-2}$ month$^{-1}$. As stated in the main text, there were no significant effects. Portions of these data have been published previously (Groffman et al. 2006; Groffman et al. 2018).

315

[Figure]

**Figure S9. Soil CH₄ sink for four vegetation zones:** (a) SpruceFir, (b) high-elevation hardwoods, (c) mid-elevation hardwoods, and (d) 320 low-elevation hardwoods. Data are shown for the three chambers per watershed in each forest type at each time. As stated in the main text, the only significant effects were for low- and high-elevation hardwoods. Portions of these data have been published previously (Groffman et al. 2006; Ni and Groffman 2018).

[Figure]

325

**Figure S10.  Cumulative N₂O sink for four vegetation zones:** (a) SpruceFir, (b) high-elevation hardwoods, (c) mid-elevation hardwoods, and (d) low-elevation hardwoods. As stated in the main text, there were no significant effects.

330

[Figure]

**Figure S11. Cumulative CH₄ sink for four vegetation zones:** (a) SpruceFir, (b) high-elevation hardwoods, (c) mid-elevation hardwoods, and (d) low-elevation hardwoods. As stated in the main text, the only significant effects were for high- and low-elevation hardwoods. Note
335   that these effects differ: treated low-elevation hardwoods are a greater sink for CH₄ but the opposite is true for high-elevation hardwoods.

[Figure]

**Figure S12. Basal area fraction for seven forest types in the treated watershed** (Battles et al. 2015b, 2015a; Driscoll et al. 2015; Driscoll Jr et al. 2015). (a) In addition to the four forest types where we have measured greenhouse gas fluxes (SpruceFir, and Low, Mid and High elevation hardwoods), a few additional categories appear in the vegetation survey data for the treated watershed. The most widespread of these is "Poor Hardwoods", which are found to the east of the Mid and High elevation hardwoods. They were divided into "High" and "Low" elevation in the 2006 and 2011 vegetation inventories (Battles et al. 2015a, 2015b); these are depicted as Poor Mid HW and Poor High HW above and their sum = Poor HW. (b) The watershed is divided into 25x25m plots on a slope-corrected grid and we can assign these plots to the categories shown based on the elevation designations in the datafiles. Maps for 2001, 2006, and 2011 are similar, except that some plots near plot 144 have no elevation designation recorded even though they were surveyed.

[Figure]

**Figure S13. Mean whole-watershed greenhouse gas savings:** (a) soil respiration and (b) $CH_4$ sink. Here we show different possibilities for combining the vegetation zones. We found no evidence for statistically significant differences in $N_2O$ emissions. We propagated the errors using standard techniques (sum of squares of standard errors).

[Figure]

**Figure S14. Whole-catchment mean cumulative (a) soil respiration and (b) soil $CH_4$ sink for comparison with Fig. 3 (main text) and Fig. S11.** Y-axis scales match Fig. 4 and Fig. S8. In these figures, we have calculated all emissions using vegetation distributions in the treated watershed. We have not used vegetation distributions from the reference watershed for our reference curves because we were primarily interested in the probable emissions from our treated watershed, had it not been treated. The reference watershed has a different elevation range and distribution of vegetation zones. Propagated standard errors from the individual chambers are shown for each point.

**4 Details of datasets used in this study**

Here we list DOIs, filenames, package IDs and access dates of our datasets. As of the time of writing these are freely available with the exception of the latest strontium isotope data. Contact J. Blum for access to those data.

Longitudinal streamwater chemistry data, treated watershed, accessed spring 2018

DOI: 10.6073/pasta/fcfa498c5562ee55f6e84d7588a980d2 (Driscoll 2016a)

File: w1long_strmchem.txt  Package ID: knb-lter-hbr.156.5  (*Uploaded 2016-03-01*)

Longitudinal streamwater chemistry data, reference watershed, accessed spring 2018

DOI: 10.6073/pasta/0033e820ff0e6a055382d4548dc5c90c (Driscoll 2016b)

File: w6long_strmchem.txt Package ID: knb-lter-hbr.127.7  (*Uploaded 2016-03-01*)

Streamwater Sr isotope and Ca/Sr, treated watershed (1997–2009, subset of our extended dataset)

DOI: 10.6073/pasta/43ebc0f959780cfc30b7ad53cc4a3d3e  (Blum 2019)

File: w1_stream_isotopes.txt  Package ID: knb-lter-hbr.139.3  (*Uploaded 2019-01-10*)

Extended data provided by J. Blum spring 2018.

Precipitation chemistry (rain/snow), treated watershed, accessed spring 2018

DOI: 10.6073/pasta/df90f97d15c28daeb7620b29e2384bb9 (Likens 2016a)

File: w1_precip_chem.txt  Package ID: knb-lter-hbr.15.9  (*Uploaded 2016-02-18*)

Precipitation chemistry (rain/snow), reference watershed, accessed spring 2018

DOI: 10.6073/pasta/8d2d88dc718b6c5a2183cd88aae26fb1) (Likens 2016b)

File: w6_precip_chem.txt  Package ID: knb-lter-hbr.20.9  (*Uploaded 2016-02-18*)

Daily streamflow, accessed spring 2018

DOI: 10.6073/pasta/727ee240e0b1e10c92fa28641bedb0a3 (Campbell 2015)

File: swd_all.txt  Package ID: knb-lter-hbr.2.6  (*Uploaded 2016-02-01*)

Daily rain/snow precipitation, accessed spring 2018

DOI: 10.6073/pasta/a84c4ecb82573486e4d080d392fe64b1 (Campbell 2016a)

File: pwd_all.txt  Package ID: knb-lter-hbr.14.8  (*Uploaded 2016-03-01*)

Daily mean temperature, accessed spring 2018

DOI: 10.6073/pasta/75b416d670de920c5ace92f8f3182964 (Campbell 2016b)

File: tdm_all.txt  Package ID: knb-lter-hbr.58.7  (*Uploaded 2016-03-03*)

Soil respiration, nitrous oxide flux and methane flux, accessed summer 2018

DOI: 10.6073/pasta/9d017f1a32cba6788d968dc03632ee03 (Groffman 2016)

File: trace_gas.txt  Package ID: knb-lter-hbr.116.10  (*Uploaded 2016-02-29*)

400

Vegetation survey data, treated watershed, 1996, accessed July 2019

DOI: 10.6073/pasta/9ff720ba22aef2b40fc5d9a7b374aa52) (Driscoll Jr et al. 2015)

File: w1_1996veg.txt  Package ID: knb-lter-hbr.40.7  (*Uploaded 2019-01-09*)

405    Vegetation survey data, treated watershed, 2001, accessed July 2019

DOI: 10.6073/pasta/a2300121b6d594bbfcb3256ca1c300c8 (Driscoll et al. 2015)

File: w1_2001veg.txt  Package ID: knb-lter-hbr.41.7  (*Uploaded 2019-01-09*)

Vegetation survey data, treated watershed, 2006, accessed July 2019

410    DOI: 10.6073/pasta/37c5a5868158e87db2d30c2d62a57e14 (Battles et al. 2015a)

File: w1_2006veg.txt  Package ID: knb-lter-hbr.142.3  (*Uploaded 2019-01-10*)

Vegetation survey data, treated watershed, 2011, accessed July 2019

DOI: 10.6073/pasta/94f9084a3224c1e3e0ed38763f8dae02) (Battles et al. 2015b)

415    Filename: w1_2011veg.txt  Package ID: knb-lter-hbr.143.3  (*Uploaded 2019-01-10*)

**References**

Arthur, MA, TG Siccama, and RD Yanai. 1999. 'Calcium and magnesium in wood of northern hardwood forest species: relations to site characteristics', *Canadian Journal of Forest Research*, 29: 339–46.

Bain, DC, and JR Bacon. 1994. 'Strontium isotopes as indicators of mineral weathering in catchments', *Catena*, 22: 201–14.

Battles, John J, Charles T Driscoll Jr, Scott W Bailey, Joel D Blum, Donald C Buso, Timothy J Fahey, Melany Fisk, Peter M Groffman, CE Johnson, and GE Likens. 2015a. 'Forest Inventory of a Calcium Amended Northern Hardwood Forest: Watershed 1, 2006, Hubbard Brook Experimental Forest. Environmental Data Initiative'.

———. 2015b. 'Forest Inventory of a Calcium Amended Northern Hardwood Forest: Watershed 1, 2011, Hubbard Brook Experimental Forest. Environmental Data Initiative'.

Battles, John J, Timothy J Fahey, Charles T Driscoll Jr, Joel D Blum, and Chris E Johnson. 2014. 'Restoring soil calcium reverses forest decline', *Environmental Science & Technology Letters*, 1: 15–19.

Bauer, James E, Wei-Jun Cai, Peter A Raymond, Thomas S Bianchi, Charles S Hopkinson, and Pierre AG Regnier. 2013. 'The changing carbon cycle of the coastal ocean', *Nature*, 504: 61–70.

Bianchi, Thomas S. 2011. 'The role of terrestrially derived organic carbon in the coastal ocean: A changing paradigm and the priming effect', *Proceedings of the National Academy of Sciences*, 108: 19473–81.

Blum, Joel D. 2019. 'Streamwater Ca, Sr and $^{87}Sr/^{86}Sr$ measurements on Watershed 1 at the Hubbard Brook Experimental Forest. Environmental Data Initiative'.

Blum, Joel D, Andrea Klaue, Carmen A Nezat, Charles T Driscoll, Chris E Johnson, Thomas G Siccama, Christopher Eagar, Timothy J Fahey, and Gene E Likens. 2002. 'Mycorrhizal weathering of apatite as an important calcium source in base-poor forest ecosystems', *Nature*, 417: 729–31.

Cai, Wei-Jun. 2011. 'Estuarine and coastal ocean carbon paradox: CO2 sinks or sites of terrestrial carbon incineration?', *Annual Review of Marine Science*, 3: 123–45.

Campbell, J. 2015. 'Hubbard Brook Experimental Forest (USDA Forest Service): Daily Streamflow by Watershed, 1956–present', *Environmental Data Initiative*.

———. 2016a. 'Hubbard Brook Experimental Forest (US Forest Service): Total Daily Precipitation by Watershed, 1956–present. Environmental Data Initiative'.

———. 2016b. 'Hubbard Brook Experimental Forest (USDA Forest Service): Daily Mean Temperature Data, 1955–present. Environmental Data Initiative'.

Canfield, Donald E, Alexander N Glazer, and Paul G Falkowski. 2010. 'The evolution and future of Earth's nitrogen cycle', *Science*, 330: 192–96.

Capo, Rosemary C, Brian W Stewart, and Oliver A Chadwick. 1998. 'Strontium isotopes as tracers of ecosystem processes: theory and methods', *Geoderma*, 82: 197–225.

Chen, Fei, Daniel G MacDonald, and Robert D Hetland. 2009. 'Lateral spreading of a near-field river plume: Observations and numerical simulations', *Journal of Geophysical Research: Oceans*, 114: C07013.

Cho, Youngil, Charles T Driscoll, Chris E Johnson, Joel D Blum, and Timothy J Fahey. 2012. 'Watershed-level responses to calcium silicate treatment in a northern hardwood forest', *Ecosystems*, 15: 416–34.

Driscoll, Charles T. 2016a. 'Longitudinal Stream Chemistry at the Hubbard Brook Experimental Forest, Watershed 1, 1991–present. Environmental Data Initiative'.

———. 2016b. 'Longitudinal Stream Chemistry at the Hubbard Brook Experimental Forest, Watershed 6, 1982–present. Environmental Data Initiative'.

Driscoll, Charles T, Scott W Bailey, Joel D Blum, Donald C Buso, Christopher Eagar, Timothy J Fahey, Melany Fisk, Peter M Groffman, CE Johnson, GE Likens, Steven P Hamburg, and Thomas G Siccama. 2015. 'Forest Inventory of a Calcium Amended Northern Hardwood Forest: Watershed 1, 2001, Hubbard Brook Experimental Forest. Environmental Data Initiative'.

Driscoll Jr, Charles T, Scott W Bailey, Joel D Blum, Donald C Buso, Christopher Eagar, Timothy J Fahey, Melany Fisk, Peter M Groffman, CE Johnson, GE Likens, Steven P Hamburg, and Thomas G Siccama. 2015. 'Forest Inventory of a Calcium Amended Northern Hardwood Forest: Watershed 1, 1996, Hubbard Brook Experimental Forest. Environmental Data Initiative'.

Duvert, Clément, David E. Butman, Anne Marx, Olivier Ribolzi, and Lindsay B. Hutley. 2018. '$CO_2$ evasion along streams driven by groundwater inputs and geomorphic controls', *Nature Geoscience*, 11: 813–18.

Fahey, Timothy J, Alexis K Heinz, John J Battles, Melany C Fisk, Charles T Driscoll, Joel D Blum, and Chris E Johnson. 2016. 'Fine root biomass declined in response to restoration of soil calcium in a northern hardwood forest', *Canadian Journal of Forest Research*, 46: 738–44.

Fahey, TJ, TG Siccama, CT Driscoll, GE Likens, J Campbell, CE Johnson, JJ Battles, JD Aber, JJ Cole, and MC Fisk. 2005. 'The biogeochemistry of carbon at Hubbard Brook', *Biogeochemistry*, 75: 109–76.

Groffman, P. M., M. C. Fisk, C. T. Driscoll, G. E. Likens, T. J. Fahey, C. Eagar, and L. H. Pardo. 2006. 'Calcium additions and microbial nitrogen cycle processes in a northern hardwood forest', *Ecosystems*, 9: 1289-305.

Groffman, P.M., C.T. Discoll, J. Duran, J.L. Campbell, L. M. Christenson, T.J. Fahey, M. C. Fisk, Colin Fuss, G. E. Likens, G.M. Lovett, L. Rustad, and P. Templer. 2018. 'Nitrogen oligotrophication in northern hardwood forests', *Biogeochemistry*, 141: 123-29.

Groffman, P.M., J.P. Hardy, M. C. Fisk, J.T. Fahey, and C.T. Driscoll. 2009a. 'Climate variation and soil carbon and nitrogen cycling processes in a northern hardwood forest.', *Ecosystems*, 12: 927-43.

Groffman, Peter M, Janet P Hardy, Melany C Fisk, Timothy J Fahey, and Charles T Driscoll. 2009b. 'Climate variation and soil carbon and nitrogen cycling processes in a northern hardwood forest', *Ecosystems*, 12: 927–43.

Groffman, Peter M. . 2016. 'Forest soil: atmosphere fluxes of carbon dioxide, nitrous oxide and methane at the Hubbard Brook Experimental Forest, 1997–present. Environmental Data Initiative'.

Hartmann, Jens, and Stephan Kempe. 2008. 'What is the maximum potential for $CO_2$ sequestration by "stimulated" weathering on the global scale?', *Naturwissenschaften*, 95: 1159–64.

Hunt, CW, JE Salisbury, and D Vandemark. 2011. 'Contribution of non-carbonate anions to total alkalinity and overestimation of pCO 2 in New England and New Brunswick rivers', *Biogeosciences*, 8: 3069–76.

Johnson, Chris E, Charles T Driscoll, Joel D Blum, Timothy J Fahey, and John J Battles. 2014. 'Soil chemical dynamics after calcium silicate addition to a northern hardwood forest', *Soil Science Society of America Journal*, 78: 1458–68.

Likens, GE. 2016a. 'Chemistry of Bulk Precipitation at Hubbard Brook Experimental Forest, Watershed 1, 1963–present. Environmental Data Initiative'.

———. 2016b. 'Chemistry of Bulk Precipitation at Hubbard Brook Experimental Forest, Watershed 6, 1963–present. Environmental Data Initiative'.

Lovett, Gary M, Mary A Arthur, and Katherine F Crowley. 2016. 'Effects of calcium on the rate and extent of litter decomposition in a northern hardwood forest', *Ecosystems*, 19: 87–97.

Marinov, Irina, Anand Gnanadesikan, Jorge L Sarmiento, JR Toggweiler, M Follows, and BK Mignone. 2008. 'Impact of oceanic circulation on biological carbon storage in the ocean and atmospheric pCO2', *Global biogeochemical cycles*, 22: 123–45.

Minocha, Rakesh, Stephanie Long, Palaniswamy Thangavel, Subhash C Minocha, Christopher Eagar, and Charles T Driscoll. 2010. 'Elevation dependent sensitivity of northern hardwoods to Ca addition at Hubbard Brook Experimental Forest, NH, USA', *Forest Ecology and Management*, 260: 2115–24.

Nezat, Carmen A, Joel D Blum, and Charles T Driscoll. 2010. 'Patterns of Ca/Sr and $^{87}$Sr/$^{86}$Sr variation before and after a whole watershed CaSiO$_3$ addition at the Hubbard Brook Experimental Forest, USA', *Geochimica et Cosmochimica Acta*, 74: 3129–42.

Ni, Xiangyin, and P.M. Groffman. 2018. 'Declines in methane uptake in forest soils', *Proceedings of the National Academy of Sciences*, www.pnas.org/cgi/doi/10.1073/pnas.1807377115.

Peters, Stephen C, Joel D Blum, Charles T Driscoll, and Gene E Likens. 2004. 'Dissolution of wollastonite during the experimental manipulation of Hubbard Brook Watershed 1', *Biogeochemistry*, 67: 309–29.

Pett-Ridge, Julie C, Louis A Derry, and Jenna K Barrows. 2009. 'Ca/Sr and 87Sr/86Sr ratios as tracers of Ca and Sr cycling in the Rio Icacos watershed, Luquillo Mountains, Puerto Rico', *Chemical Geology*, 267: 32–45.

Raymond, Peter A. 2017. 'Temperature versus hydrologic controls of chemical weathering fluxes from United States forests', *Chemical Geology*, 458: 1–13.

Renforth, Phil, and Gideon Henderson. 2017. 'Assessing ocean alkalinity for carbon sequestration', *Reviews of Geophysics*, 55: 636–74.

Rosi-Marshall, Emma J, Emily S Bernhardt, Donald C Buso, Charles T Driscoll, and Gene E Likens. 2016. 'Acid rain mitigation experiment shifts a forested watershed from a net sink to a net source of nitrogen', *Proceedings of the National Academy of Sciences*, 113: 7580–83.

Salisbury, Joseph E, Douglas Vandemark, Christopher W Hunt, Janet W Campbell, Wade R McGillis, and William H McDowell. 2008. 'Seasonal observations of surface waters in two Gulf of Maine estuary-plume systems: Relationships between watershed attributes, optical measurements and surface pCO2', *Estuarine, Coastal and Shelf Science*, 77: 245–52.

Schlesinger, William H, and Ronald Amundson. 2018. 'Managing for soil carbon sequestration: Let's get realistic', *Global Change Biology*, 00: 1–4.

Seitzinger, Sybil P, and Anne E Giblin. 1996. 'Estimating denitrification in North Atlantic continental shelf sediments.' in, *Nitrogen cycling in the North Atlantic Ocean and its watersheds* (Springer).

Shao, Shuai, Charles T Driscoll, Chris E Johnson, Timothy J Fahey, John J Battles, and Joel D Blum. 2016. 'Long-term responses in soil solution and stream-water chemistry at Hubbard Brook after experimental addition of wollastonite', *Environ. Chem*, 13: 528–40.

Stumm, W, and J Morgan. 1996. *Aquatic Chemistry: Chemical Equilibria and Rates in Natural Waters*.

Virta, Robert L. 2000. 'Minerals Yearbook: Wollastonite', Unites States Geological Survey National Minerals Information Center, Accessed 11 September https://www.usgs.gov/centers/nmic/wollastonite-statistics-and-information.

Williamson, Phillip, Douglas WR Wallace, Cliff S Law, Philip W Boyd, Yves Collos, Peter Croot, Ken Denman, Ulf Riebesell, Shigenobu Takeda, and Chris Vivian. 2012. 'Ocean fertilization for geoengineering: a review of effectiveness, environmental impacts and emerging governance', *Process Safety and Environmental Protection*, 90: 475–88.

---

## Author Comment (AC2) · 5 Oct 2020

Reviewer 2: Overall, I thought this paper was excellent. The manuscript is polished, thorough, and well structured in way that presents a data-heavy study in a concise manner. Testing ways to remove carbon dioxide from the atmosphere is critical for mitigating the response to anthropogenic climate change, which makes this paper of particular significance. I only have one query which I would like some comment on in the discussion, with a couple of minor comments. I recommend that the paper is accepted after these are addressed.

Response: Thank you!

Reviewer 2: Discussion: The long term efficacy of carbon capture and storage, both in

geological and modern examples, seems to hinge on whether organic (via biomass) or inorganic (via carbonate) carbon are the dominant sinks for increases in atmospheric $CO_2$. Given that this study concludes that uptake into biomass is an important factor for carbon storage in the catchments, what does this mean for the ability of experiments such as this to function on longer time scales (i.e. >100 years). Is this a one-off procedure that can be implemented on a catchment, or can it be repeated with a minimum repose time? Will the draw down via organic and inorganic pathways change with repeated treatments perhaps? I know this is going to be speculative, but I think it would be beneficial for the authors to share their thoughts on how this may be able to be integrated into long term catchment management strategies.

Response: Yes, this would be speculative but we can discuss how this type of treatment integrates with long-term management and organic/inorganic pathways of carbon sequestration. Such a discussion would come at the very end of the discussion and would be a good way to end the paper given that Reviewer 1 recommended removing the Conclusions. Forestry practices such as harvests are known to produce changes in biogeochemical cycling and ion export in streams, potentially affecting decisions re timing of treatments and need for repeat treatments. This may be particularly pertinent for evergreen forests which may be difficult to treat from the air, but also for deciduous forests with high productivity.

Reviewer 2: Minor comments: Line 30: 71 degrees west, rather than -71 degrees east

Response: We are happy to change this (but will comply with journal guidelines if they prefer degrees East).

Reviewer 2: Line 120: Repetition of "Mohseni and Stefan"

Response: Yes, Endnote added the authors' names again. We can remove "following Mohseni and Sefan".

Reviewer 2: Line 127: What does "mm/time" mean?

Response: This means millimeters (cubic meters of water per square meter of land) per unit time. We calculated rolling annual fluxes at the sampling interval of the input chemistry data (approximately one month). We can change this to mm per year.

Reviewer 2: Line 192: Repetition of "Battles et al."

Response: Endnote can probably put in a reference without repeating the author names; alternatively we can reword the sentence to avoid the repetition.

Reviewer 2: Line 350: Replace "3.4 4" with "3.44"

Response: Thanks for catching this error, we are happy to fix it.

Reviewer 2: Line 418: A possibility for a low cost alternative to wollastonite could be volcanic ash (see e.g. Longman et al., 2020; https://doi.org/10.1016/j.ancene.2020.100264), particularly in catchments with volcanic deposits nearby

Response: We are happy to cite Longman et al in the paragraph starting on line 418, as an alternative lower-cost treatment along with mine waste. Soils located very close to volcanoes probably already contain considerable volcanic material but transport by ship may be viable.

---

## Author Response (AR2)

**Response to reviewers**

**Reviewer 1**: Summary: Taylor et al. present results from an enhanced rock weathering (ERW) field experiment in Hubbard Brook Experimental Forest in the northeastern United States. The authors show observational and modeling evidence in support of sustained carbon dioxide removal for 15 years following the application of silicate minerals to the the experimental plots in 1999. Overall, I find the observational and technical pieces of this manuscript to be very strong. I also found this manuscript difficult to read. I believe the authors could improve the readability, and likely the impact, of the manuscript by revising the structure and flow of the manuscript. Currently, there are an extensive number of equations, missing topic sentences, and redundant sections. These all need to be edited to improve the manuscript. I have tried to highlight some examples below.

**Response:** We are pleased that the reviewer found the work to be strong, and we think the manuscript has been improved as a result of these comments. In response to the comments about the greenhouse gas balance, we have revised the relevant text and figures. We are grateful that the reviewer noticed that the text of Sec. 3.3 had been copied into Sec. 3.5; the correct text has now been reinstated. Section 2 has been reorganised, and therefore the subsection and equation numbers have changed. The first four equations have been redesignated R1 through R4 following journal guidelines for chemical reactions. The number of equations in Methods has been reduced and these are now numbered 1 through 7. Introductory text has also been added to the beginnings of Sections 2.2 and 2.3 which should help orient readers to the topics to be addressed.

**Reviewer 1: General comments:** Please streamline Secs. 2.2.3-2.2.5. After reading them many times, it is still very confusing which equations were used in the modeling, and which are there merely for background context.

**Response**: These sections tell readers exactly how we calculate the key variable giving rise to the ERW concept ($CO_2$ consumption), so we were reluctant to relegate any of this material to supplementary information. However, we had overlooked the possibility of appendices at the end of the main manuscript. Our former sections 2.2.4 and 2.2.5 provided useful and even critical detail for experts but could be skipped by non-experts without compromising our main conclusions, and as such that material now forms our new Appendices A and B. Our former Eq. 6 "Non-Wo-$CO_2$" has been removed, and we considered moving the "$CO_{2,ions}$" equation to an appendix. However, it is appropriate to retain it in the main text  because it is a very well-known, commonly-used equation in the $CO_2$ consumption literature and results from it do appear in Table 3 for comparison with bicarbonate-derived $CO_2$ consumption.  The two equations and reworded text related to total catchment $CO_2$ consumption remains in Section 2.2.3, while the two equations giving our upper and lower limits for the treatment effect on $CO_2$ consumption are now in Section 2.2.4 with some of the accompanying text reworded.  Our former Section 2.2.6 is now renumbered 2.2.5 accordingly.

**Reviewer 1:** Please use math fonts to better differentiate between text and equations. It is very hard to follow the train of logic in the manuscript, which employs 16 equations, without appropriate fonts.

**Response:** According to the *Biogeosciences* guidelines, mathematical variables (other than chemical species) should be displayed in italics and we had not done this.  We also converted the equations and variables where they appear in the text to the Cambria Math font and put them in boldface to make them easier to see.  The number of equations is reduced to a total of seven excluding the four overall chemical reactions in Section 1; the latter are now renumbered R1 through R4 following journal guidelines.

**Reviewer 1:** This manuscript would benefit from a table listing all model variables presented in the manuscript with descriptions and units. It is too difficult to keep track of all variables, especially without the use of Math font currently. Also, all model variables need to be used consistently throughout the manuscript. For instance, "X" is used in Eq. 9 and "X_Ca" is used in Eq. 10.

**Response**: We have created such a table (now our Table 1) listing the variables in (renumbered) Equations 1 through 7 which includes the variable names, units, the sections where they are discussed, equations where they appear, figures and tables where they can be seen, and descriptions.  This table is at the beginning of the Methods following a small paragraph of introductory text. "X" is now "$X_{Ca}$" everywhere.

**Reviewer 1**: Please embed figures and tables in the appropriate positions in the manuscript, not at the end. This greatly facilitates comprehension of the non-typeset document by reviewers.

**Response**: Done.

**Reviewer 1**: Please make arrangements to make the Matlab scripts publicly available, via Github, as a series of supplemental files to the manuscript, or through some other appropriate means. Doing so improves the reproducibility of the science, and allows others to access them without needing to make a "request" (as indicated in the manuscript).

**Response**: The MATLAB scripts have been prepared for dissemination and are now included along with a README file in a supplementary .zip file.

**Reviewer 1**: This manuscript is missing a study site figure (probably as Fig. 1) that orients readers to the HBEF and the study and control watersheds.

**Response**: We have added a new Figure 1 showing the study site and locations of our streamwater and trace gas samples.

**Reviewer 1**: Please do not reference equations that have yet to be presented in the manuscript (e.g. Eq. 13, L128).

**Response**: Spurious reference to Eq. 13 removed from Sec. 2.2.3.

**Reviewer 1: Specific comments:**

**Reviewer 1**: There are two Sec 2.1.1: Site description and Treatment description. Please correct.

**Response:** Corrected, see line 90 of the updated manuscript. Yes, the second one should be Sec. 2.1.2 Treatment description

**Reviewer 1**: Sec 2.1.1 (Site description): Watershed W1 is never introduced. It needs to be introduced here prior to mention of its flow rates (L74)

**Response**: Watershed 1 had been introduced after watershed 6, but we have now reversed this so watershed 1 is introduced first. See lines 74 and 75 of the updated manuscript.

**Reviewer 1**: I find the transition between Secs. 2.2 and 2.3 to be difficult to follow. Sec. 2.2 presents the modeling approach and the first sentence in Sec 2.3 begins talking about wood production. Please provide some introductory material in Sec. 2.3 prior to discussing the details of the GHG calculations

**Response**: We have moved and revised the material which had previously formed Sec. 2.3.4 (Greenhouse gas budget for a treatment) to the beginning of Sec. 2.3, as it introduces the greenhouse gas balance and the variables to be discussed in subsequent subsections. We updated the text to make the description easier to understand. See Section 2.3.1 of the revised manuscript.

**Reviewer 1:** Inline calculations (e.g. L195) are very difficult to follow and hinder comprehension. Please consider alternative ways to deliver this information to readers.

**Response:** We removed the inline calculation and revised the text to make it easier for readers to see how the values in the former Table 2 (now Table 4) were calculated. See Sec. 2.3.2 of the revised manuscript.

**Reviewer 1**: Secs. 3.2 and 3.3 are essentially sensitivity analyses of the model to different assumptions or scenarios. As such, I think Figs. 2 and 3 could be placed in the SI in order to keep the main figures focused solely on the observational results of the ERW experiment (or model results of the observations)

**Response**: Agreed. We have moved those two figures to the supplementary information, which has been reorganised to accommodate them.

**Reviewer 1**: Sec. 3.5 is identical to Sec. 3.3 (unless I am missing something). Please remove.

**Response:** We are grateful to the reviewer for catching this mistake. Sec. 3.5 is where the logistical penalties and greenhouse gas balance results are discussed but the text was inadvertently deleted during the final stages of editing. It has now been reinstated.

**Reviewer 1**: Sec 5: This section does not add any new information to the manuscript, especially since key findings were reviewed in Sec. 4. Please remove.

**Response**: Agreed. The Conclusions have been deleted.

**Reviewer 1**:  Figs. 1 and 2 are too small to be easily readable. Please enlarge.

**Response**: The old Fig. 1 (now Fig. 2) and has redrawn to improve use of space and legibility. The original Fig. 2 is now in supp. info, and has had white space removed to look larger.

**Reviewer 1**: Figs. 1-3: The dashed lines representing the treatment should be identified in the figure captions.

**Respons**e: Done.

**Reviewer 1**: Fig. 4: Is time-integrated CO2 flux shown on the y-axis (as implied by L 324)? If so, please correct the y-axis label accordingly.

**Response**: The Y axis label now says "Cumulative flux".

**Reviewer 1**: Table 2/Fig. 5: I find the terminology and axis references incredibly confusing. Please use alternate language that more clearly indicates whether the total greenhouse gas budget has increased or decreased.

**Response**: The introductory text of Sec. 3.1, the relevant table (now Table 4) and Fig. 5 have been revised and simplified. Figure 5 has been broken up so that the logistical penalties now form our new Fig. 4. The new Fig. 5 now has only one panel. It shows the variable names appearing in the old Table 2 (now Table 4) and the old Equations 15 and 16 (now just Eq. 6), as well as the meanings of those variables so that the figure is comprehensible without reading the text.

**Reviewer 1**: Fig. 5: missing panel captions (e.g., "a)", "b)", etc.). Also the caption is excessively long and needs to be shortened.

**Response**: There were several unnecessary sentences in the caption, but as stated above our old Fig 5a and Fig 5b now form separate new Fig. 4 and 5 (with Fig 5 overhauled as noted above), each with the relevant part of the old caption. Panels c and d of the original figure added little to the story and they have been removed along with their references in the caption.

**Reviewer 1**: Fig. 5: I could not find a reference to this figure in the text. Please add a reference.

**Response**: The references to this figure are in the reinstated text of Sec. 3.5.

**Reviewer 1**: Figure axis text needs to be enlarged across all figures.

**Response**: Main manuscript figures now have larger text.

**Reviewer 2**

Morgan Jones (Referee) m.t.jones@geo.uio.no

**Reviewer 2**: Overall, I thought this paper was excellent. The manuscript is polished, thorough, and well structured in way that presents a data-heavy study in a concise manner. Testing ways to remove carbon dioxide from the atmosphere is critical for mitigating the response to anthropogenic climate change, which makes this paper of particular significance. I only have one query which I would like some comment on in the discussion, with a couple of minor comments. I recommend that the paper is accepted after these are addressed.

Thank you for this positive assessment of our work.

**Reviewer 2**: Discussion: The long term efficacy of carbon capture and storage, both in geological and modern examples, seems to hinge on whether organic (via biomass) or inorganic (via carbonate) carbon are the dominant sinks for increases in atmospheric CO2. Given that this study concludes that uptake into biomass is an important factor for carbon storage in the catchments, what does this mean for the ability of experiments such as this to function on longer time scales (i.e. >100 years). Is this a one-off procedure that can be implemented on a catchment, or can it be repeated with a minimum repose time? Will the draw down via organic and inorganic pathways change with repeated treatments perhaps? I know this is going to be speculative, but I think it would be beneficial for the authors to share their thoughts on how this may be able to be integrated into long term catchment management strategies.

**Response:** We have added a paragraph at the end of the discussion (now the last paragraph of the paper as Reviewer 1 recommended removing the Conclusions). The new paragraph (starting on Line 533 of the revised manuscript) discusses how treatments can integrate with long-term management and organic/inorganic pathways of carbon sequestration.

**Reviewer 2**: Minor comments: Line 30: 71 degrees west, rather than -71 degrees east

**Response**: Done. See Line 30 of the revised manuscript.

**Reviewer 2**: Line 120: Repetition of "Mohseni and Stefan"

**Response**: Yes, Endnote added the authors' names again. We removed "following Mohseni and Stefan". See Line 141 of the revised manuscript.

**Reviewer 2**: Line 127: What does "mm/time" mean?

**Response**: This means millimeters (cubic meters of water per square meter of land) per unit time. We calculated rolling annual fluxes at the sampling interval of the input chemistry data (approximately one month).  We changed this to mm per year. See Line 148 of the revised manuscript.

**Reviewer 2**: Line 192: Repetition of "Battles et al."

**Response**: We reworded the text to avoid the repetition.  See lines 227 and 2228 of the updated manuscript.

**Reviewer 2:** Line 350: Replace "3.4 4" with "3.44"

**Response**: Done. See Line 445 of the revised manuscript.

**Reviewer 2**: Line 418: A possibility for a low cost alternative to wollastonite could be volcanic ash (see e.g. Longman et al., 2020; https://doi.org/10.1016/j.ancene.2020.100264), particularly in catchments with volcanic deposits nearby

**Response**: We now cite Longman et al (2020) on line 527 of the revised manuscript.

[revised manuscript text omitted]

**2.2 Geochemical modelling and $CO_2$ consumption fluxes**

$CO_2$ consumption, the CDR pathway most closely associated with ERW, can be calculated from concentrations of either bicarbonate or the base cations released during weathering (Eq. R1). These two approaches may provide different answers if bicarbonate is reduced in the presence of other acids (Eqs. R2–R4). To calculate bicarbonate-derived $CO_2$ consumption, we must model the speciation of dissolved inorganic carbon (DIC). This depends on two variables which must also be modelled because we do not have a time series: streamwater $pCO_2$ and streamwater temperature. We then calculate total catchment $CO_2$ consumption fluxes and treatment effects, taking care to account for differences in sampling frequency between chemistry samples and water flow measurements.

**Commented [LT7]:** Treated and references watersheds introduced (Reviewer 1)

**Commented [LT8]:** Fixed subsection number (Reviewer 1)

**Commented [LT9]:** Section renamed for clarity as section 2.2 does not cover the statistical modelling of the watershed gas fluxes, and new introductory paragraph summarising sec. 2.2 so that readers know exactly what to expect (Reviewer 1)

[revised manuscript text omitted]

**Commented [LT13]:** Mathematics font (Reviewer 1, subsequent instances highlighted)

**Commented [LT14]:** Time units specified (Reviewer 2)

**Commented [LT15]:** Old Eq 6 "Non-Wo-CO2" removed, and text for CO2,ions moved and slightly tweaked so that the two most common expressions for total catchment CO2 consumption are presented together.

**Commented [LT16]:** It seemed sensible to give the treatment effects their own subsection. The material from the old subsections 2.2.4, 2.2.5 covering rainwater corrections and strontium isotopes have been relegated to appendices.

**Commented [LT17]:** Old figure 2 moved to Supp. Info. (Reviewer 1)

**Commented [LT18]:** X is now XCa everywhere and in mathematics font (Reviewer 1)

**Commented [LT19]:** X is now XCa everywhere and in mathematics font (Reviewer 1)

**Commented [LT20]:** X is now XCa everywhere and in mathematics font (Reviewer 1)

(Wo-Ca) determines the $CO_2$ consumption associated with the HBEF wollastonite treatment.  Our optimistic treatment effect
based on calcium rather than bicarbonate is:

$$Wo\text{-}CO_{2,Ca}(t) = 2 \times X_{Ca} \times [Ca^{2+}](t) \times flow(t). \tag{4}$$

Equations (3) and (4), together with our flux calculations accounting for sparsity of concentration data compared to daily flow
data (Sec. 2.2.5), should help avoid major uncertainties in catchment-scale $CO_2$ consumption calculations: the provenance of
the cations and variations in concentration and discharge (Moon et al., 2014).

**2.2.5 Flux calculations**

To ensure that fluxes from our two watersheds were comparable and to correct for the sparsity of solute measurements
compared to flow measurements, we created rolling annual flow-adjusted fluxes using Method 5 of Littlewood et al. (1998) at
five evenly-spaced points each year:

$$Flux = scale \times \left[\frac{\sum_{i=1}^{M} C_i Q_i}{\sum_{i=1}^{M} Q_i}\right] \times \left[\frac{\sum_{k=1}^{N} Q_k}{N}\right], \tag{5}$$

where $Q_i$ is the measured instantaneous stream flow, $C_i$ is the concentration for sample $i$, $M$ is the number of streamwater
chemistry samples in the year (usually 12), $Q_k$ is the $k^{th}$ flow measurement, and $N$ is the number of flow measurements. In
our case, daily flow measurements (Campbell, 2015) and ~monthly streamwater samples (Driscoll, 2016b, a) were available.
Therefore, the mean concentration for the preceding twelve months is multiplied by the mean flow for the same period, suitably
scaled to get the total annual flux. Without sub-daily timestamps for the longitudinal streamwater chemistry data, we used
daily total flows rather than instantaneous flows. Tests suggested that there was little difference between using mean daily
instantaneous flows and the mean daily total flows.

**2.3 Greenhouse gas balance**

The success of any treatment for climate change mitigation is determined by the net greenhouse gas ($CO_2$ equivalent) fluxes
prior to and following treatment, at the treatment site and downstream.  In addition to increased $CO_2$ consumption, desireable
outcomes for a treatment include increased ecosystem carbon storage in biomass and soils, and decreases in ecosystem,
downstream and logistical greenhouse gas emissions.

**2.3.1 Greenhouse gas budget for the wollastonite treatment**

At the HBEF, we have measured the $CO_2$ consumption due to the wollastonite treatment in two different ways and these
determine our range of values to be incorporated in our GHG budget.  Several other treatment effects can be estimated relative
to the reference watershed, but some aspects of the total GHG balance are missing.  For example, we have measurements of

**Commented [LT21]:** X is now XCa everywhere and in mathematics font (Reviewer 1)

**Commented [LT22]:** Renumbered subsection

**Commented [LT23]:** Here we fixed another instance of a reference with the author names duplicated (Reviewer 2 caught several other instances).

**Commented [LT24]:** Mathematics fonts in renumbered equation and subsequent text (Reviewer 1)

**Commented [LT25]:** This section is reorganised, and the sections have been renumbered (Reviewer 1)

**Commented [LT26]:** Introductory material added, some taken and rewritten from the old section 2.3.4 (Reviewer 1)

soil respiration (root+heterotrophic) and dissolved organic carbon (DOC) export in streamwater, but we lack measurements of canopy respiration from leaves and stems, and export of particulate organic carbon in streamwater. Our partial greenhouse gas budget for the HBEF wollastonite treatment will therefore be given by

$$\Delta GHG = \Delta wood + \Delta SRESP + \Delta CH4 + \Delta N2O + \Delta CONS + \Delta NO3N2O + \Delta DOC + LOGPEN. \tag{6}$$

where our partial GHG treatment effect ($\Delta GHG$) is the sum of greenhouse gas sink and source responses. Measured sinks for the wollastonite experiment include biomass in wood ($\Delta wood$), $CO_2$ consumption ($\Delta CONS$), and a soil sink for methane ($\Delta CH4$). Sources include $N_2O$ emissions both from soil ($\Delta N2O$) and exported nitrate ($\Delta NO3N2O$), and $CO_2$ emissions from soil respiration ($\Delta SRESP$), exported dissolved organic carbon ($\Delta DOC$), and logistical operations ($LOGPEN$).

Sink effects are defined as positive if the sink increases and are given by the difference (treated−reference) between the two watersheds, whereas source effects are defined as positive for reductions in greenhouse gas emissions (reference−treated). With these definitions, penalties are negative and reduce $\Delta GHG$ in Eq. (6). Logistical emissions and $CO_2$ consumption due to weathering of applied wollastonite are zero for the reference watershed, so we expect $LOGPEN$ to be negative and $\Delta CONS$ to be positive.

Wood is a longer-term carbon sink than leaves or twigs so we have chosen to let this represent our biomass increment. Eq. (6) neglects ecosystem disturbances including fire, and possible carbonate mineral precipitation in soils. There is no evidence for the latter at the HBEF.

We used a range of emissions factors for $N_2O$ to estimate the penalty associated with nitrate export ($\Delta NO3N2O$); low: 0.0017 kg$N_2$O-N kg$^{-1}$ DIN (Hu et al., 2016) and high: 0.0075 kg$N_2$O-N kg$^{-1}$ DIN (De Klein et al., 2006), where DIN is dissolved inorganic nitrogen dominated by nitrate. This $N_2O$ was then converted to $CO_{2e}$ ($CO_2$ equivalents in terms of cumulative radiative forcing) given the 100-year time horizon global warming potential (Pachauri et al., 2014) (GWP$_{100}$) for $N_2O$: 265 g$CO_{2e}$ g$^{-1}$ $N_2O$. Likewise, $\Delta CH4$ was converted to $CO_{2e}$ ($CO_2$ equivalents in terms of cumulative radiative forcing) given GWP$_{100}$ for $CH_4$: 28 g$CO_{2e}$ g$^{-1}$ $CH_4$.

**2.3.2 Carbon sequestration in wood**

We calculate our treatment effect on wood production as the difference between the treated and reference watershed mean wood production (Battles et al., 2014) over two five-year periods. We considered these differences (treated−reference) to be an estimate of the treatment effect on potentially long-term (decades to centuries) biomass carbon sequestration. Assuming 46.5% of the woody biomass is carbon (Martin et al., 2018), our calculated cumulative additional C sequestration in the treated watershed over ten years was 20.7 mol C m$^{-2}$ (9.1 t $CO_2$ ha$^{-1}$). Our optimistic and pessimistic values are derived from the 95% confidence intervals for the five-year mean values (Battles et al., 2014).

**Commented [LT27]:** The two equations (15 and 16) from the old section 2.3.4 have been removed and replaced with a single equation for clarity (Reviewer 1).

**Commented [LT28]:** Equation renumbered following relegation of subsections to appendices (Reviewer 1)

**Commented [LT29]:** Equation renumbered

**Commented [LT30]:** Section renumbered, reworded and inline equation removed (Reviewer 1)

**232 2.3.3 Greenhouse gas emissions from soils**

Measurements (Groffman, 2016) were taken at four elevations in the treated watershed and at points just west of the reference watershed starting in 2002 (**Fig. 1**). Gas samples were collected from chambers placed on three permanent PVC rings at each of these eight sites (Groffman, 2016). The data were not normally distributed so were analyzed with Kruskal-Wallis tests at the 0.05 significance level; however, tests with one-way ANOVA produced the same overall results. All analyses were done in Matlab R2016a.

Cumulative curves for each of the 24 chambers were generated by matching the dates of the measurements, excluding points which were missing data for any chamber and allowing up to a week's discrepancy between catchments. Nearly all discrepancies were within one day. Assuming diurnal variation was minor compared to seasonal variation, each datum (g C

$m^{-2}$ $hour^{-1}$) was multiplied by 24 hours and by 30 days to get gC $m^{-2}$ $month^{-1}$. There was no extrapolation to fill gaps in the dataset; results are internally consistent but not comparable to other datasets. We were particularly interested in the elevation- specific responses, as the different elevations have distinct tree species compositions and below-ground responses to the wollastonite treatment (Fahey et al., 2016).

The HBEF experimental watersheds are divided into 25×25m plots on slope-corrected grids. Vegetation has been surveyed four times since the late 1990s and assigned a zone designation in each plot (Driscoll et al., 2015;Driscoll Jr et al.,

2015;Battles et al., 2015b, a) (Fig. S12). To estimate the respiration savings over the whole watershed, we added the areas of individual plots which were assigned to our four vegetation types (Low, Mid and High hardwoods, and Spruce-Fir). Because there were seven vegetation types in the datasets, we compared all types with pairwise Kruskal-Wallis tests at the 0.05

significance level using the basal area data for the six dominant tree species. Kruskal-Wallis tests were appropriate because the data, and therefore the differences from the means (residuals), were not normally distributed. These tests suggested that the "extra" vegetation types ("Birch/Fern Glade", and "Poor Hardwoods" at High and Mid elevations) could be combined with

Spruce-Fir, High and Mid Hardwoods respectively. Watershed fractions for our combined forest types were 0.155 for

SpruceFir, 0.16 for High Hardwoods, 0.415 for Mid Hardwoods, and 0.27 for Low Hardwoods. When creating our composite treatment effects for the entire watershed, we considered a treatment effect to be present only where our statistical analyses suggested significantly different fluxes.

**257 2.3.4 Logistical carbon emissions costs**

We used the 1999 upstate New York $CO_2$ emission factor for electricity generation from oil (United States Environmental

Protection Agency, 1999) (0.9 Mg $CO_2$ $MWh^{-1}$), and rearranged Equation 28 of Stamboliadis (Stamboliadis et al., 2009):

$$e_p = \frac{e^{\left[\frac{(\ln s/\alpha)}{\mu}\right]}}{3600 \times 1000},$$  (7)

where the specific surface area $s$ (1600 $m^2$ $kg^{-1}$ for our treatment) is related to the specific potential energy $e_p$ of the material (kJ $kg^{-1}$), with theoretical parameters (Stamboliadis et al., 2009) $\alpha$=139 $m^2$ $kJ^{-1}$ and $\mu$=0.469 (dimensionless). We convert this

**Commented [LT31]:** Section renumbered and reference to the new site figure added

**Commented [LT32]:** The old Fig 3 was moved to supp. info and we have removed a reference to it. (Reviewer 1)

**Commented [LT33]:** Supp. info figures have been renumbered.

**Commented [LT34]:** Subsection renumbered (Reviewer 1)

**Commented [LT35]:** Equation renumbered following reorganisation, mathematics font used (Reviewer 1)

[revised manuscript text omitted]

**3 Results**

**3.1 Wollastonite treatment increased streamwater CO₂ export**

**Commented [LT38]:** This section has some small edits for clarity (eg "treated" and "REF" rather than "T" and "R" and removal of "stream"), use of mathematics fonts and renumbering of the figure from 1 to 2 following addition of the new site figure (Reviewer 1)

[Figure]

**Figure 2: Inorganic CO₂ capture at the Hubbard Brook Experimental Forest.** (a) Observed calcium and (b) calcium export in the reference (grey) and treated (blue) watersheds along with the contribution from sources other than wollastonite (red) and the time of treatment (vertical dotted line). (c) Calculated CO₂ consumption due to the treatment (*Wo-CO₂,Ca*, Eq. 9). (d) Modelled streamwater bicarbonate, (e) CO₂ consumption (*CO₂,HCO3*, Eq. 5), and (f) CO₂ consumption due to the treatment (*Wo-CO₂,HCO3*, Eq. 7), colours as for calcium. Simulations (d–f) account for the presence of organic acids (+OA). All calcium export (b) and CO₂ consumption curves (c,e,f) were calculated with flow-normalised concentrations and corrected for sparsity of samples (Methods).

**Commented [LT39]:** Figure renumbered, excess whitespace removed so it looks larger, treatment time (vertical dotted line) now appears in the legend, and mathematics font used when referring to model variables.

[revised manuscript text omitted]

**Commented [LT41]:** Former Fig. 3 relegated to supp info, table renumbered and within the section, and mathematics fonts for model variables in text and table (Reviewer 1)

**Commented [LT42]:** Table renumbered and within section, mathematics fonts used for model variables listed in the new Table 1 (Reviewer 1)

**Commented [LT43]:** Missing reference for the Mauna Loa data added

**Commented [LT44]:** Three erroneous values replaced with correct values

**3.4 Amplification of organic carbon sequestration by wollastonite treatment**

In reversing long-term $Ca^{2+}$ depletion of soils, the silicate rock treatment significantly increased forest growth and wood production between 2–12 years post-treatment relative to the reference watershed (Battles et al., 2014). This forest response increased total carbon sequestration by 20.7 mol C $m^{-2}$ or 9.1 t $CO_2$ $ha^{-1}$ during those ten years as a result of the treatment (Methods).

Changes in greenhouse gas (GHG) emissions from soils represent a further route to affecting the climate mitigation potential of the wollastonite treatment. Despite a rapid increase of one pH unit in the upper organic soil horizon (Oie), soil respiration $CO_2$ fluxes showed no significant difference between watersheds during the first three years after treatment (Groffman et al., 2006). However, our analysis of newly available longer-term datasets indicates that the treatment significantly reduced soil respiration in the high elevation hardwood zone (~660–845m a.s.l.) ($\chi^2(1,270)$=17.2, $P < 0.001$), possibly due to reduced fine-root biomass (Fahey et al., 2016) rather than changes in microbial activity (Groffman et al., 2006). No significant effects on soil respiration were detected in any of the other HBEF vegetation zones (**Fig. 3**). The wollastonite treatment increased the soil sink strength for $CH_4$ ($\chi^2(1,266)$=30.8, $P < 0.001$) in the low-elevation hardwood zone (482–565m a.s.l.), while it decreased in the high elevation zone ($\chi^2(1,268)$=22.3, $P < 0.001$) (SI Appendix, Fig. S8). There were no significant treatment effects on soil $N_2O$ fluxes in any vegetation zone (SI Appendix).

**Commented [LT45]:** Figure renumbered following removal of two figures and addition of the new site figure (Reviewer 1)

[Figure]

**Figure 3:** **Long-term soil respiration responses to wollastonite treatment at Hubbard Brook Experimental Forest.** Cumulative soil $CO_2$ respiration responses of treated and untreated (a) high elevation hardwoods, (b) high elevation conifers, (c) low elevation hardwoods or (d) mid-elevation hardwoods. Plots show cumulative means ± 1 SE for three chamber measurements at each site and time. Reference data were collected from untreated forests immediately adjacent to the western edge of our reference catchment. *P*-values from Kruskal-Wallis tests comparing treated and reference raw data (SI Appendix) are shown.

**3.5 Logistical $CO_2$ emissions and net CDR**

We next considered carbon emissions (penalties) for logistical operations involved in mining, grinding, transporting and applying the wollastonite (**Fig. 4**, **Table 2**). In the HBEF experiment, wollastonite was mined and milled on site near Gouverneur, New York. We used $CO_2$ emissions factors for electricity generation in upstate New York (United States Environmental Protection Agency, 1999) to estimate the maximum $CO_2$ penalty for mining and grinding to the mean particle

Commented [LT46]: Figure renumbered and placed within section (Reviewer 1)

Commented [LT47]: Duplicate text (sec 3.3) removed and replaced with correct text. The entire section is highlighted as neither reviewer has seen it as yet. (Reviewer 1)

Commented [LT48]: Reference to Fig 4, formerly top left panel of the old Fig 5 (Reviewer 1)

size 16 μm diameter (Methods).  However, local hydropower (Energy Information Administration, 1997) and regional nuclear power suggest these costs could have been zero.  This would represent a substantial carbon saving for the overall ERW process relative to prior expectation of ERW studies in which grinding $CO_2$ emissions account for up to 30% reduction in ERW-CDR

efficiency (Renforth, 2012;Moosdorf et al., 2014).

[Figure]

**Figure 4: Carbon penalties for the wollastonite treatment.** Carbon penalties for logistic elements of the treatment are compared with literature estimates for large-scale rollout of enhanced rock weathering for the HBEF treatment (3.44 t ha[-1]), with and without long-distance transport for pelletization.

In the HBEF experiment, the milled wollastonite was transported by highway to Allerton, Illinois, for pelletization and then returned to the staging area near Woodstock, New Hampshire (round trip >3150 km). Transportation $CO_2$ emissions were 0.22–0.61 t $CO_2$ t $Wo^{-1}$.  Given coal power in central Illinois, we estimate pelletization emitted up to 0.02 t $CO_2$ t $Wo^{-1}$

(Methods).  Application at Hubbard Brook occurred via 55 ~5-km helicopter flights, which gives a further $CO_2$ cost of 0.01–

0.15 t $CO_2$ t $Wo^{-1}$.  In total, these logistical operations emitted 0.23–0.69 t $CO_2$ t $Wo^{-1}$, or 0.8–2.4 t $CO_2$ ha[-1] for the 11.8 ha of the HBEF treated watershed (**Table 4**).  However, local pelletization could have reduced heavy duty vehicle (HDV) transport distance to ~400 km and lowered total $CO_2$ emitted during logistical operations to 0.04–0.15 t $CO_2$ t $Wo^{-1}$. At other forested sites, where wind-drift of material is not critical, pelletization may not be necessary.

**Commented [LT49]:** This was the top left panel of the old Figure 5.  It is now a figure on its own and within the section text (Reviewer 1)

**Table 4.** Measured elements of the treatment effect on the greenhouse gas budget for the Hubbard Brook Experimental Forest
wollastonite experiment.

**Commented [LT50]:** Table renumbered, simplified and within section text. (Reviewer 1)

| Equation 14 | Greenhouse gas sinks[a] and emissions[a] (t $CO_{2e}$ ha$^{-2}$) | Pessimistic | Optimistic |
|---|---|---|---|
| **Ecosystem responses[b]** | | | |
| *Δwood* | Wood production sink increased over ten years[c] | 8.946 | 9.542 |
| *ΔSRESP* | Soil respiratory $CO_2$ emissions have reduced[a] since 2002 | 2.213 | 2.646 |
| *ΔCH4* | Soil methane sink has increased since 2002 | 0.015 | 0.029 |
| *ΔN2O* | Soil $N_2O$ emissions since 2002 (no significant difference) | 0 | 0 |
| | **Net ecosystem response at the treatment site through 2014** | **11.174** | **12.218** |
| **Downstream sequestration and emissions responses** | | | |
| *ΔCONS* | $CO_2$ consumption sink through 2014 (***Wo-CO$_{2,HCO3}$*** and ***Wo-CO$_{2,Ca}$***) | 0.025 | 0.129 |
| *ΔNO3N2O* | Downstream $N_2O$ emissions[d] from treatment date through 2014 | –0.071 | –0.016 |
| *ΔDOC* | DOC export emissions[d,e] from treatment date through 2014 | –0.203 | 0 |
| | **Net downstream balance through 2014** | **–0.228** | **-0.129** |
| **Logistics:** | | | |
| | Mining/Grinding given hydro or nuclear/petroleum power | –0.162 | 0 |
| | Helicopter (~55 5-km flights) | –0.051 | –0.021 |
| | HDV transport (New York to Illinois to New Hampshire) | –2.135 | –0.787 |
| | Pelletization (in Illinois, coal power) | –0.068 | 0 |
| *LOGPEN* | **Total logistical emissions** | **–2.416** | **–0.808** |
| *ΔGHG* | **Partial treatment effect on greenhouse gas balance** | **8.509** | **11.523** |

[a]Defined as the difference between watersheds: treated–reference for sinks and reference–treated for emissions

[b]Some possible treatment responses such as canopy respiration and particulate organic carbon export are unknown.

[c]After Battles et al. (2014). We have not attempted to extrapolate these results.

[d]ΔDOC and ΔNO3N2O are penalties because these lead to $CO_2$ and $N_2O$ emissions downstream.

[e]The "optimistic" value for DOC assumes complete burial and undesireable low oxygen conditions in downstream waters.

These carbon emission penalties must be subtracted from watershed carbon removal to calculate net CDR for the
wollastonite treatment at HBEF (**Fig. 5**; **Table 4**). Compared in this way, we find increased wood production over ten years
(Battles et al., 2014) repays the total logistical $CO_2$ costs 4–12 times over. The components (**Fig. 5; Table 4**) comprise 8.5–
11.5 t$CO_2$ ha$^{-1}$ of the total GHG budget associated with the wollastonite treatment (Methods). These figures would increase
to 10.4–12.2 t$CO_2$ ha$^{-1}$ if the wollastonite had been pelletized anywhere along the route from Gouverneur to New Hampshire.

**Commented [LT51]:** Reference to Fig. 5 (Reviewer 1)

Wollastonite treatment effects on streamwater chemistry play a minor role in the greenhouse gas budget (**Fig. 5;**
**Table 4**). For our hypothetical ten-fold higher treatment (34.4 t ha$^{-1}$), $CO_2$ consumption calculated by assumed calcium release
is ~10 times higher, but carbon emission penalties scale with increased rock mass. Assuming pelletization near the mine to
reduce transport costs, the total logistical penalty would be 1.2–5.1 tCO$_2$ ha$^{-1}$. In total, net CDR would be 6.8–12.4 tCO$_2$ ha$^{-1}$
for the the ten-fold larger treatment if none of the other GHG fluxes changed. We have not attempted to extrapolate other forest
biomass and soil GHG fluxes or streamwater DOC and NO$_3^-$ responses.

[Figure]

**Figure 5: Carbon responses for the wollastonite treatment.** Elements of the greenhouse gas balance associated with the wollastonite
treatment (**Table 4**). The CO$_2$ consumption range is given by ***Wo-CO$_{2,HCO3}$*** calculated by Eq. (3) and ***Wo-CO$_{2,Ca}$*** calculated by Eq. (4), time-
integrated from the application date through 2014. Nitrate export in streamwater leading to N$_2$O greenhouse gas emissions downstream and
a small increase in the soil CH$_4$ sink have been converted to CO$_2$-equivalents (Methods). Exported DOC is assumed to be respired
downstream.

**Commented [LT52]:** This new figure replaces the old Figure 5 top right panel (the old bottom panels added little to the story and have been removed). The Y axis labels are the same variable names appearing in the new Eqn 14 and in the revised and renumbered Table 3. Each item also has short explanatory text explaining what those variables are. Following removal of three panels the caption is also shorter (Reviewer 1).

[revised manuscript text omitted]

**Commented [LT58]:** Discussion of ERW integrated with forestry practices (Reviewer 2)

**Commented [LT59]:** Conclusions removed (Reviewer 1)

**Commented [LT60]:** Old section 2.2.5 material moved to Appendices as it describes details useful for experts in the field but which may disrupt the flow of the main text (Reviewer 1)

**Commented [LT61]:** Mathematics fonts and rewording in this section which is retained with Eqns 11/12 as they describe a key correction necessary to calculate CO2 consumption (Reviewer 1)

**Commented [LT62]:** Typo corrected: hyphen replaced by en-dash.

**Commented [LT63]:** "however" added for readability.

**Appendix B   Fraction of calcium derived from wollastonite**

We applied an existing two-component mixing model (Peters et al., 2004):

$$X_{Ca}(t) = \left[ \frac{\left(\left(\frac{^{87}Sr}{^{86}Sr}\right)_{post} - \left(\frac{^{87}Sr}{^{86}Sr}\right)_{pre}\right)\left(\frac{Sr}{Ca}\right)_{pre}}{\left(\left(\frac{^{87}Sr}{^{86}Sr}\right)_{post} - \left(\frac{^{87}Sr}{^{86}Sr}\right)_{pre}\right)\left(\frac{Sr}{Ca}\right)_{pre} + \left(\left(\frac{^{87}Sr}{^{86}Sr}\right)_{Wo} - \left(\frac{^{87}Sr}{^{86}Sr}\right)_{pre}\right)\left(\frac{Sr}{Ca}\right)_{Wo}} \right], \tag{B1}$$

where pre-app and post-app refer to pre-application and post-application streamwater concentrations and Wo refers to wollastonite. The Sr data (Blum, 2019) have been extended through 2015 (Fig. S1a). See Supplementary Information for further discussion of the use of strontium and its isotopes as tracers of $Ca^{2+}$ provenance.

**Code availability**

The aqueous geochemistry software PHREEQC software, along with documentation, is freely available from the USGS website (https://www.usgs.gov/software/phreeqc-version-3). MATLAB® may be purchased from the MathWorks website (https://uk.mathworks.com/products/matlab.html). Our MATLAB code and scripts used for this project are provided in a supplementary .zip file, without guarantees that these will run with MATLAB versions other than R2016a or on non-Linux operating systems.

**Data availability**

Our data are available from the Long Term Ecological Research (LTER) Network Data Portal. This public repository can be accessed via the Hubbard Brook Ecosystem Study website: https://hubbardbrook.org/d/hubbard-brook-data-catalog

See Supplement for a full list of filenames, package IDs, DOIs and access dates.

**Author contributions**

All authors contributed to project conceptualization and interpretation of model results. L.L.T. undertook model simulations and data analysis. L.L.T. and D.J.B. drafted the manuscript with edits and revisions from all authors. C.T.D. designed the wollastonite watershed study, provided data and observations for model simulations. J.D.B. provided strontium isotope datasets. P.M.G. provided soil respiration, nitrous oxide and methane flux data.

**Commented [LT64]:** Material from old section 2.2.4 moved to Appendices as it describes details useful for experts in the field but which may disrupt the flow of the main text (Reviewer 1)

**Commented [LT65]:** Mathematics fonts, XCa, and slight rewording for clarity. (Reviewer 1)

**Commented [LT66]:** Supp. Info figures have been renumbered following addition of the original Figs 2 and 3 from the main manuscript (Reviewer 1)

**Commented [LT67]:** Code has been cleaned up for clarity and made available in a new supplementary .zip file (Reviewer 1)

**Competing interests**

The authors declare that they have no conflict of interest.

**Disclaimer**

**Acknowledgements**

L.L.T. and D.J.B. gratefully acknowledge funding from the Leverhulme Trust through a Leverhulme Research Centre Award
(RC-2015-029). This manuscript is a contribution of the Hubbard Brook Ecosystem Study. Hubbard Brook is part of the Long-
Term Ecological Research (LTER) network, which is supported by the National Science Foundation (DEB-1633026). L.L.T.
thanks Ruth Yanai for a helpful discussion about vegetation, Fred Worrall for advice on flow adjustment and flux calculation,
Peter Wade for advice on the initial PHREEQC setup and Andrew Beckerman and Evan DeLucia for constructive criticism
and advice on statistical modelling. We are grateful to Gregory Lawrence for information about applying lime treatments to
the Appalachian Trail corridor, to Lisa Martel for providing the locations of the trace gas sampling sites and to Habibollah
Fahkraei for creating the watershed map with weir and trace gas sampling locations in Fig. 1. We are grateful for comments
from W. Brian Whalley and editor Tyler Cyronak, and for reviews from Morgan Jones and an anonymous referee which led
to major improvements in this manuscript.

> **Commented [LT68]:** New site figure (Fig. 1, Reviewer 1)

> **Commented [LT69]:** Editor: we had hitherto omitted to thank our reviewers. Brian's comments on our revised manuscript led to repositioning of the table of model variables requested by Reviewer 1.

[revised manuscript text omitted]

> **Commented [LT70]:** All figures and tables moved to sections referring to them (Reviewer 1)